# The EBS-A* algorithm: An improved A* algorithm for path planning

**Huanwei Wang**, **Shangjie Lou, Jing Jing\*, Yisen Wang, Wei Liu, Tieming Liu**

State Key Laboratory of Mathematical Engineering and Advanced Computing, Zhengzhou, China

\* jingjing_cs@hotmail.com

## Abstract

Path planning plays an essential role in mobile robot navigation, and the A* algorithm is one of the best-known path planning algorithms. However, the traditional A* algorithm has some limitations, such as slow planning speed, close to obstacles. In this paper, we propose an improved A*-based algorithm, called the EBS-A* algorithm, that introduces expansion distance, bidirectional search, and smoothing into path planning. The expansion distance means keeping an extra space from obstacles to improve path reliability by avoiding collisions. Bidirectional search is a strategy searching path from the start node and the goal node simultaneously. Smoothing improves path robustness by reducing the number of right-angle turns. In addition, simulation tests for the EBS-A* algorithm are performed, and the effectiveness of the proposed algorithm is verified by transferring it to a robot operating system (ROS). The experimental results show that compared with the traditional A* algorithm, the proposed algorithm improves the path planning efficiency by 278% and reduces the number of critical nodes by 91.89% and the number of right-angle turns by 100%.

**Data Availability Statement:** Relevant data are available from Github: https://github.com/wanghw1003/EBAStar.

**Funding:** The author(s) received no specific funding for this work.

## Introduction

Mobile robots are a comprehensive system that integrates environmental perception, dynamic decision-making, behavior control, task planning, and execution. This accomplishes functions such as movement, automatic navigation, multisensor control, and network interaction. Mobile robots can be widely used in stations, airports, and post offices, and are rapidly commandeering important roles in our daily lives. Robots must plan an appropriate path to move when navigating in a complex or uncertain actual environment. Path planning determines a collision-free path in a given environment and has been developed in mobile robots in the last few decades.

Over the past decades, different path planning methods [1–5] have been proposed and demonstrated on robots or automated guided vehicles (AGVs) with various applications. Classical path planning algorithms include genetic algorithm (GA) [6, 7], ant colony optimization (ACO) algorithm [8–10], rapidly-exploring random trees (RRT) algorithm [11, 12], and A-Star (A*) algorithm [13–15]. The A* algorithm is based on graph searching and is one of the most commonly used path planning methods. In the A* algorithm, the optimal path is generated by convergence.

**Competing interests:** The authors have declared that no competing interests exist.

The performance of the $A^*$ algorithm is mainly reflected in the speed of path planning and the robustness of the planned path. Although scholars have performed much research on the $A^*$ algorithm, there are still some defects, such as small distances between the path and obstacles and slow speed due to right-angle turns. These factors lead to decreasing robustness of the planned path. The speed of path planning and path smoothness are other issues under research that determine the efficiency of the algorithm and the speed of the mobile robot. To improve the robustness of the conventional $A^*$ algorithm, expansion distance and path smoothing are introduced into the proposed algorithm. The expansion distance means keeping an extra space from obstacles to improve path reliability by avoiding collisions, for the expanded nodes are no longer traversed, the speed of path planning is improved. The expansion distance is equivalent to reducing the map scale in some sense. Smoothing improves path reliability by reducing the number of right-angle turns. To improve the speed of path planning, bidirectional search strategy is introduced into the proposed algorithm. This strategy searches from the start node and the goal node simultaneously.

The main contributions of the research in this paper are as follows: we propose three methods for the improvements of the conventional $A^*$ algorithm, including expansion distance, bidirectional search, and smoothing. We introduced these methods into $A^*$ algorithm form a new algorithm named EBS-$A^*$. In addition, simulation tests are performed, and the results show that compared with the traditional $A^*$ algorithm and other variants ($A^*$ with expansion distance, bidirectional search $A^*$ and geometric $A^*$ [16], the EBS-$A^*$ algorithm achieves better performance with respect to path robustness and path planning speed. In addition, to test its effectiveness, the EBS-$A^*$ algorithm is transplanted into the FS-AIROBOTB mobile robot hardware platform, produced by China Huaqing Yuanjian, and tested in the real world.

The rest of this manuscript is organized as follows. Section 2 introduces the research performed on the $A^*$ algorithm in recent years. In section 3, the basic theory of the $A^*$ algorithm is introduced. In section 4, the three optimization strategies of the $A^*$ algorithm, and the pseudo-code of the EBS-$A^*$ algorithm are given. The time complexity analysis of the EBS-$A^*$ algorithm is introduced in section 5. In section 6, the EBS-$A^*$ algorithm is tested and compared by simulation. In section 7, the EBS-$A^*$ algorithm is tested in the robot operating system (ROS) and its performance is verified in the actual environment. Last, conclusions are drawn in section 8.

## Related work

Path planning algorithms include several classification methods, which are differentiated based upon available environmental knowledge. There are multiple classification methods for path planning algorithms. For example, groups include graph search-based algorithms, including Dijkstra algorithm, State Lattice algorithm, etc. sampling-based algorithms, including RRT, etc. Additionally, various research works propose that path planning algorithms can actually be described as classical search algorithms and heuristic search algorithms. Classical algorithms include depth-first search (DFS), breadth-first search (BFS), and Dijkstra algorithm. These algorithms are path planning algorithms based on graph search. Heuristic algorithms include $A^*$ algorithm, $D^*$ algorithm, GA algorithm, ACO algorithm, Artificial Neural Network (ANN) algorithm, and Simulated Annealing (SA) algorithm.

Path planning algorithms are also divided into global path planning and local path planning based upon available environmental knowledge. Global path planning seeks the optimal path given largely complete environmental information and is best performed when the environment is static and perfectly known to the robot. Therefore, global path planning is also called static path planning. By contrast, local path planning is most typically performed in unknown or dynamic environments, and local path planning is also called dynamic path planning.

For application scenarios such as warehousing and logistics, path planning in a static environment assumes that the robot perceives the environment and uses local path planning algorithms when the environmental information is not fully grasped. A* is used for shortest path evaluation based on the information regarding the obstacles present in the static environment [17]. The shortest path evaluation for the known static environment is a two-level problem. The problem comprises a selection of feasible node pairs and the shortest path evaluation based on the obtained feasible node pairs [18]. Neither of the abovementioned criteria are available in a dynamic environment, which makes the algorithm inefficient and impractical in dynamic environments. In this case, the dynamic path planning algorithm is not suitable for use in a static environment. Classical algorithms include D* algorithm, Artificial Potential Field algorithm. A* algorithm is chosen because it represents the foundational algorithms used within contemporary real-time path planning solutions in a static environment. Novel research builds on this algorithm to achieve additional performance and efficiency.

Dijkstra algorithm relies upon a greedy strategy for path planning. It is used to find the shortest path in a graph. It is concerned with the shortest path solution without formal attention to the pragmatism of the solution [17]. Such algorithms also include BFS and DFS. The most prominent disadvantages of these algorithms are that they require traversing the map completely, which results in a large amount of calculation, low efficiency and weak collision-free. To overcome Dijkstra's computational intensity when conducting blind searches, A* [19] and its variants are presented as state of the art algorithms for use within static environments. The A* algorithm can plan the shortest path in the map, but it needs to traverse around the path nodes and select the minimum path cost. Therefore, the algorithm necessitates a large amount of calculation and long calculation time, and the efficiency of the algorithm will decrease with the expansion of the map scale [20]. Due to the characteristics of the graph search algorithm itself, the A* algorithm uses rasterized maps as the map representation method, which makes its path smoothness poor and results in excessive right-angle turns, resulting in reduced reliability.

The heuristic function guides the A* algorithm to search for the shortest path. The heuristic function of the traditional A* algorithm uses Manhattan distance [21], Euclidean distance [22] and Diagonal distance [23], which are designed in subsequent research. The algorithm plans the path by employing the heuristic function to calculate the path cost.

There have been many research achievements with respect to the A* algorithm, and we focus on the efficiency and robustness of the algorithm. Scholars have carried out extensive research on the efficiency optimization of the A* algorithm. An A* optimization method is proposed [24] to solve the efficiency problem of the algorithm through two improvements. First, the evaluation function is weighted to enhance the reliability of the heuristic function. Second, a node-set centered on a certain point is constructed in the rasterized maps. When the node set contains obstacle nodes, this node is marked as an "untrusted point" and will not be searched. These optimizations are applied to improve the efficiency of the A* algorithm, however, the method increases the calculation requirements of the algorithm. Scholars focus on the storage of nodes to optimize the A* storage method [25]. The storage method accesses the array element by searching the sequence number of the node through only one search, while the traditional A* algorithm needs to traverse multiple nodes to complete the search process. The method is very limited in improving the efficiency of the algorithm, it only optimizes the A* algorithm program. A method called time efficiency A* was proposed to improve the efficiency of the algorithm [26]. This method calculates the cost by the heuristic function before the collision phase instead of the initialization phase; therefore, the algorithm can effectively reduce the runtime. This method only optimizes the opportunity to calculate the value of the

heuristic function and does not optimize the heuristic function or decrease the number of search nodes.

The A* optimization methods introduced above only incompletely optimize the efficiency of the algorithm and still present defects, which means that the efficiency of the algorithm can still be more thoroughly optimized. None of the above algorithms consider optimizing the robustness of the algorithm. Therefore, the improvements and variants of the A* algorithm remain flawed.

A constrained A* method is proposed for unmanned surface vehicles (USVs) in a maritime environment [21], and the method with a USV enclosed by a circular boundary is a safety distance constraint on the generation of optimal waypoints to avoid collisions. This reflects the collision-free performance of the method. The concept of "safety distance" is also proposed in this research, but the method includes a USV enclosed by a circular boundary, which means that the USV is surrounded and does not account for obstacles of the maps.

The path is not smooth enough to dynamically avoid collision, which is another obvious shortcoming of the traditional A* algorithm. To overcome this problem, a global path planning method that perceives the characteristics of the local environment is proposed [27]. This method employs the A* algorithm to plan the global optimal path in a known static environment, deletes redundant nodes, and then generates local sequence nodes on the deleted global path to optimize the global path. This method guarantees the performance of the A* algorithm when it is used in a dynamic environment. To overcome the collision issue, smoothing is one of the most well-known methods. The traditional A* algorithm plans some sharp turns and causes some problems for mobile robots. A smoothing of the A* algorithm is introduced in [28]. The smoothed A* algorithm [29] generates path and redundant waypoints by using cubic spline interpolation and three path smoothers. Smoothing is one of the most effective methods to reduce right-angle turns and reduce the risk of collision. Several A* optimization algorithms were used for comparison, evaluation, and application scenario selection in [13], including several modifications (Basic Theta*, Phi*) and improvements (RSR, JPS). A Hybrid A* algorithm is proposed and improves the traditional A* algorithm when employed in autonomous vehicles. This method can plan the shortest possible path in a hybrid environment for a vehicle.

The existing improved A* algorithm only optimizes either for efficiency or for robustness. However, there is no existing algorithm aiming at comprehensive performance. Path planning algorithm plays an essential role in the autonomous navigation of mobile robots. Since mobile robots are widely used in the real world, it is very necessary to propose an improved A* algorithm with strong robustness and high efficiency. It has huge potential application and commercial value in the industrial field.

## Basic theory of traditional A* algorithm

The A* algorithm was first proposed and described in detail in [13], and it is one of the best-known path planning algorithms. A* algorithm is a heuristic search algorithm, which aims to find a path from the start node to the goal node with the smallest cost by searching among all possible paths. Heuristic information related to the characteristics of the problem is utilized to guide its performance, so it is superior to other blind search algorithms [30]. Due to the heuristic search algorithm, the A* algorithm consists of an Open table, a Closed table, and a heuristic function and employs the heuristic function to evaluate the distance from an arbitrary node to the goal node on a 2D plane. The A* algorithm is defined as a best-first algorithm because each cell in the configuration space is evaluated according to the value.

$$f(n) = g(n) + h(n) \tag{1}$$

where $g(n)$ is the cost of the path from the start node to the current node $n$, $h(n)$ is the cost of the path from node $n$ to the goal node through the selected sequence of nodes, and $h(n)$ is the heuristic function of the A* algorithm. $f(n)$ is denoted as the evaluation function of node $n$. This sequence ends in the actually evaluated node. Each adjacent node of the actually reached node is evaluated by $f(n)$. The node with the minimum value of $f(n)$ is chosen as the next node in the sequence. The advantage of the A* algorithm is that other distances can be adopted, modified, or added as standard distances.

## Proposed method

In this section, three optimization strategies are proposed to improve the efficiency and robustness of the A* algorithm.

### Expansion distance

The path generated by the traditional A* algorithm may be very close to the obstacle. If the path is adopted by a mobile robot as the planning path, the mobile robot will have a high risk of collision with obstacles. Therefore, it is indispensable to consider maintaining an appropriate distance from obstacles during path planning. The appropriate distance is the concept proposed in this manuscript: expansion distance.

The expansion distance means keeping an extra space around the obstacles as the safe distance during path planning. Robots adapt rasterized maps as path planning maps. Expansion distance adopts a grid as the basic unit to expand outward around obstacles. Expansion distance is the shortest distance that the path is allowed to approach obstacles. The value of the expansion distance is determined by parameters such as the speed of the robot, the size of the robot model, and the number of grids.

The nodes of expanded distance will not be visited together with the obstacles during the path planning. Expansion distance will be used as a "collision buffer" between the robot and the obstacles, which can effectively reduce the collision risk of the robot during the travel process. Therefore, expansion distance can increase the robustness of the algorithm. The superiority of the expansion distance is not limited to enhancing the robustness of the algorithm, it is also effective in improving the efficiency of the algorithm. Since the expanded nodes are no longer visited by the algorithm, expansion distance is equivalent to reducing the map scale in some sense. The total number of nodes that the algorithm needs to traverse is reduced acordingly. Therefore, expansion distance can improve the efficiency of the algorithm. The schematic diagram of expansion distance is shown in Fig 1.

Regarding the selection of the expansion distance size, it is generally selected as a grid of the rasterized map by default. In the simulation test of this manuscript, we chose a grid as the size of expansion distance. When the robot radar constructs a real environmental map, the robot is usually modeled as a cylinder or a sphere. Normally, the expansion distance defaults to the radius of the cylinder or sphere as the expansion distance in the real environment. This distance can not only ensure the reliability of the path but also ensure acceptably minimal waste of the traveling space. The bilateral expansion distances are the size of a robot itself when there are obstacles on both sides.

How is the expansion distance automatically decided for different environments? This is a question we must consider. We refer to the robot equivalent model in the robot operating system to discuss the expansion distance. In this manuscript, the following assumptions of the robot and obstacles are made to simplify the model.

The robot model is equivalent to a cylinder in ROS, with a radius of $r$ and cruising speed $V_r$, which satisfies $V_r \leq V_{max}$, where $V_{max}$ is the maximum cruising speed determined by the

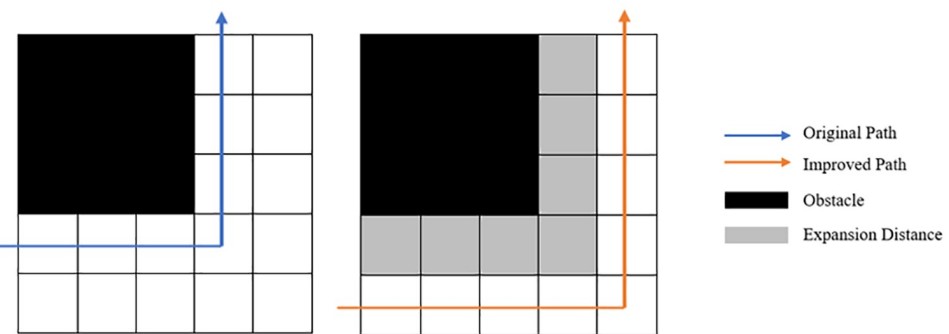

**Fig 1. Schematic diagram of expansion distance.**

performance of the robot. $V_r$ is a speed threshold, $V_i$ is the current speed of the robot. When $V_i \leq V_r$, the expansion distance is only expanded by one node, and when the current speed is greater than $V_r$, the probability of a collision between the robot and an obstacle increases. In this case, the number of expanded nodes should increase. The obstacle is equivalent to one grid or more square grids. The mapping rule between the robot model and the map is that the robot radius is equal to the length of a grid. The mapping rule between the number of expansion nodes and the speed is that the ratio of the current speed and the speed threshold.

$$E(V_i) = \begin{cases} r & V_i \leq V \\ \dfrac{V_i}{V_r}r & V_r < V_i \leq V_{max} \end{cases} \tag{2}$$

where $E(V_i)$ is the number of expansion nodes. Regarding the selection of the expansion distance magnitude, when $V_i \leq V_r$, the expansion distance defaults to the radius of the cylinder as the expansion distance. This distance provides a sufficient "collision buffer" for the reliability of the path, and ensure acceptably minimal waste of the physical space the robot travels. The bilateral expansion distances are the size of a robot itself when there are obstacles on both sides. As the speed of the robot increases, the risk of robot collisions will increase. Correspondingly, only expanding the expansion distance can ensure that the robustness of the algorithm does not decrease. When $V_i$ increases, $E(V_i)$ should increase accordingly, otherwise the risk of robot collision will increase. Therefore, the determination of the expansion distance follows a linear relationship with the speed $V_i$.

## Bidirectional search optimization

The A* algorithm is a path planning algorithm based on graph search, which is developed from BFS. Bidirectional search optimization strategy is a method to complete path search by traversing nodes in a grid map, so it is reasonable to use the bidirectional search method to improve the A* algorithm.

BFS is a blind search method and is a traversal algorithm of connected graphs. The purpose of BFS is to check all nodes in the graph systematically. In other words, BFS does not consider the possible location of the result and searches the entire graph thoroughly until it finds the result. The basic search process is that BFS starts from the root node and traverses the nodes of the tree (graph) along the width of the tree (graph). If all nodes are visited, the algorithm stops. The queue data structure is generally used to assist in the realization of the BFS algorithm. The search process of BFS is shown in Fig 2.

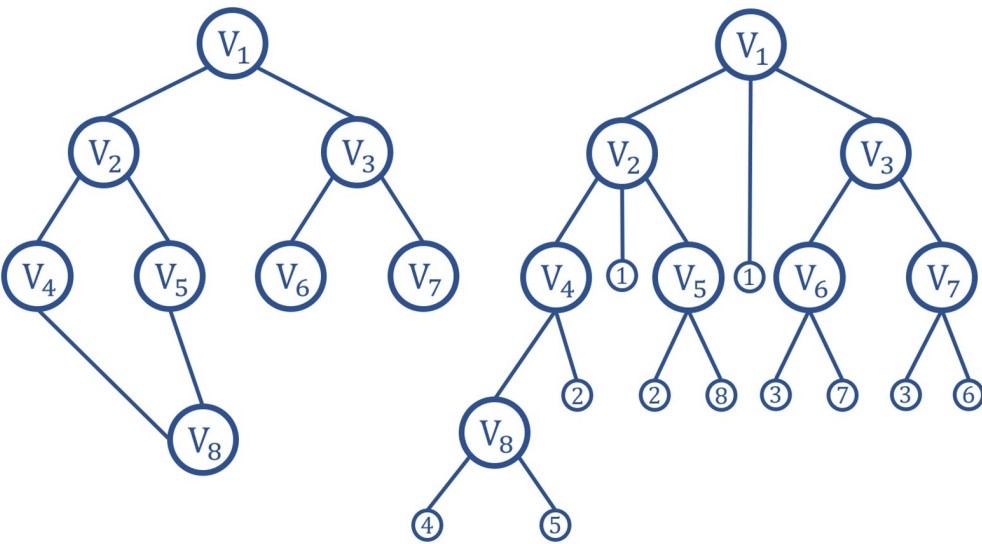

**Fig 2. The process of BFS.**

The search process of the traditional A* algorithm is unidirectional search, and the search process occur from the start node to the end node. The bidirectional search method introduced is performed in this manuscript. Bidirectional search simultaneously searches from the start node to the end node and from the end node to the start node. When an obstacle is encountered during the search, the algorithm still searches the path until the intersection according to the established search mode. When the forward and reverse search nodes are adjacent nodes during the search process, the search process finishes. The positive and negative incomplete paths are spliced together to form a complete collision-free path. A schematic diagram of the search optimization is shown in Fig 3. The left figure is the unidirectional search, and the right figure is the bidirectional search.

The search method introduces a parallel idea and searches from the start node and the goal node at the same time. A function call completes the two searches of the start node and the end node, which reduces the number of function calls and improves the path planning efficiency of the algorithm. In addition, the bidirectional search reduces the number of traversed nodes compared with unidirectional search, which further accelerates the path planning efficiency.

In schematic diagrams, the green node represents the start node, the red node represents the end node, the blue node represents the next search node, the yellow node represents the forward searched node, the gray node represents the reverse searched node, and the purple node represents the common node to be searched.

The traditional graph search algorithms are not considered the features of the path planning problem, like DFS and BFS. The path is searched by the strategy set beforehand for any problem, and the search process will not be optimized according to the features of the problem. The A* algorithm is developed based on the BFS algorithm. The concept of heuristic is introduced on the basis of the BFS algorithm. The heuristic information is obtained according to the features of the problem, which will guide the search in the optimal direction. Such as speeding up the search process and improving efficiency. The traditional A* algorithm uses Manhattan distance as its heuristic equation. Manhattan distance is defined as follows:

$$h(n) = |x_a - x_b| + |y_a - y_b| \qquad (3)$$

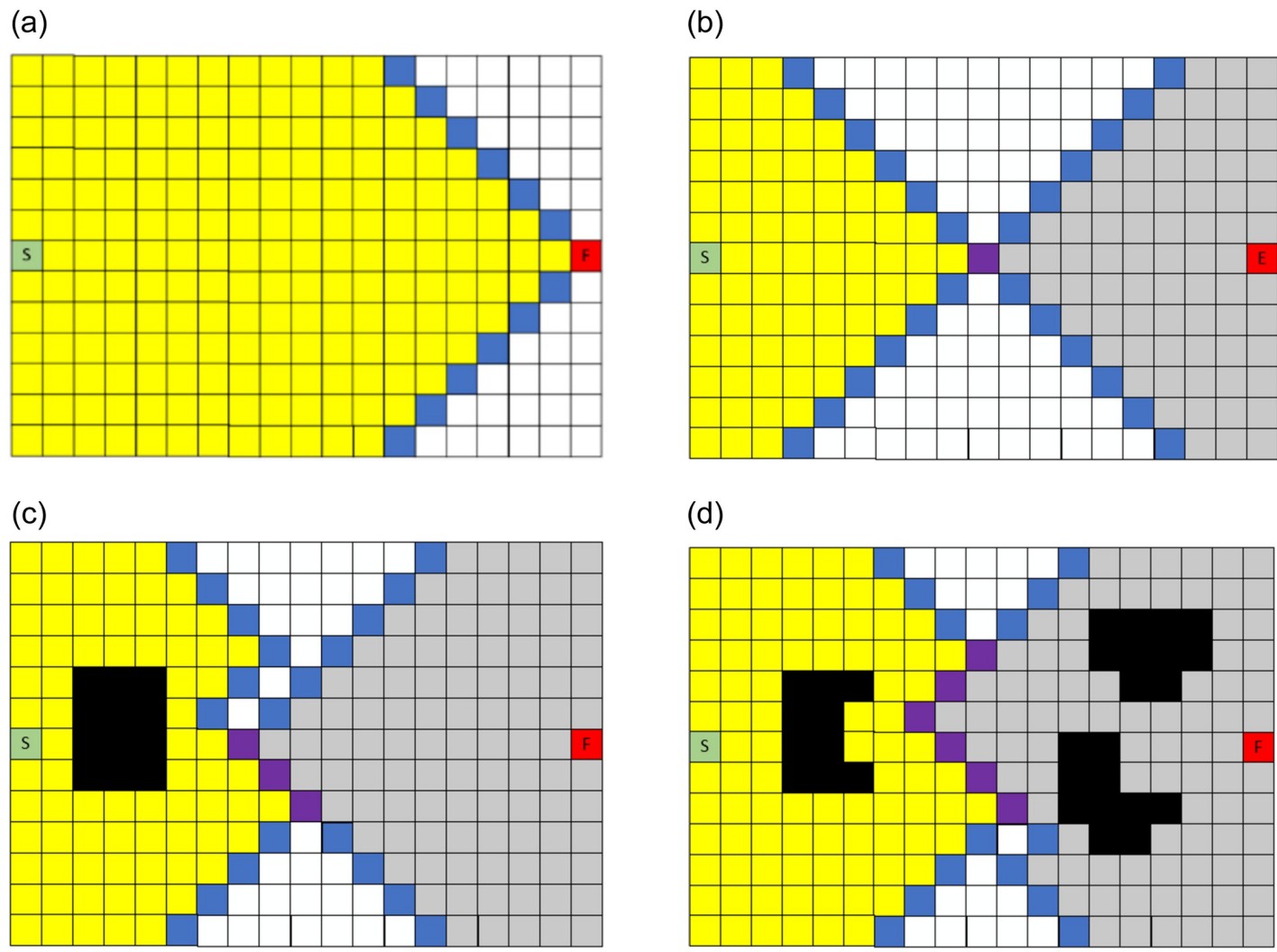

**Fig 3. Schematic diagrams of bidirectional search.** (a) Unidirectional search. (b) Bidirectional search without obs. (c) Bidirectional search with an obstacle. (d) Bidirectional search with multiple obstacles.

$h(n)$ is the heuristic function, $(x_a, y_a)$ is the coordinates of the goal node, $(x_b, y_b)$ is the coordinates of any node.

## Smoothing optimization

The path of the traditional A* algorithm usually consists of a series of nodes and many polyline segments connected to them. There are three primary disadvantages regarding the path [29]. First, the goal of the A* algorithm is to find the minimum path length cost in such a way that the generated jags are not minimized. It sacrifices turning costs to obtain the shortest path. Second, the planned path is not continuous. Third, due to the existence of right-angle turns, the mobile robot needs to decelerate sharply during movement, which affects the speed and path robustness. Therefore, to overcome these shortcomings, the path of the conventional A* algorithm needs to be smoothed.

Bezier curve is a space curve and has good geometric properties, which is proposed by the French engineer Pierre Bezier in 1962. Bezier curve is one of the methods used to smooth the

path, it has been widely used in computer graphics and computer-aided design. If the control point of the Bezier curve is a convex polygon, that is, the feature polygon is convex, the Bezier curve is also convex, which is one of the advantages of the Bezier curve. Unlike other types of curves, such as cubic splines or polynomials, Bezier curve does not pass through all the data points used to define it. The points used to define Bezier curve are called control points. Polygons that can be drawn from these control points are called Bezier polygons. The turning points are the points where the slope of the curve changes its sign. Bezier curves have fewer turning points so that it is smoother than cubic splines.

Bezier curve has the following properties:

1. Symmetry, the $i^{th}$ coefficient of the curve is the same as the reciprocal $i^{th}$ coefficient.

2. Convex hull properties, Bezier curve is always contained in the convex hull of the polygon defined by all control points.

3. End-points properties, The first control point and the last control point on the curve are exactly the start point and the end point of the Bezier curve.

4. Recursion, which means that the coefficient of the Bezier curve satisfies the following formula.

$$B_{i,n}(t) = (1-t)B_{i,n-1}(t) + tB_{i-1,n-1}(t), \{i = 0, 1, ..., n\} \tag{4}$$

The radius of curvature of the Bezier curve varies smoothly from the starting point to the endpoint because of its continuous higher order derivatives. A Bezier curve of degree n is a parametric curve composed of Bernstein basis polynomials of degree $n$ and it can be defined as:

$$P(t) = \sum_{i=1}^{n} p_i B_{i,n}(t), t \in [0, 1] \tag{5}$$

Where $t$ indicates the normalized time variable, $P_i(x_i, y_i)^T$ represents the coordinate vector of the $i^{th}$ control point with $x_i$ and $y_i$ being the components corresponding to the X and Y coordinate, respectively, $B_{i,n}$ is the Bernestein basis polynomials, which represents the base function in the expression of Bezier curve, and it is defined as follows:

$$B_{i,n}(t) = C_n^i t^i = \frac{n!}{i!(n-i)!} t^i (1-t)^{n-i}, i = 0, 1, ..., n. \tag{6}$$

The derivatives of Bezier curve are determined by the control points, and the first derivative of a Bezier curve in formula 5. is expressed as in formula 7. Moreover, higher-order derivatives of a Bezier curve can also be calculated.

$$\dot{P}(t) = \frac{dP(t)}{dt} = n \sum_{i=0}^{n-1} B_{i,n-1}(t)(P_{i+1} - P_i) \tag{7}$$

In the two-dimensional space, the curvature of a Bezier curve with respect to t is expressed as follows:

$$k(t) = \frac{1}{R(t)} = \frac{\dot{P}_x(t)\ddot{P}_y(t) - \dot{P}_y(t)\ddot{P}_x(t)}{\left(\dot{P}_x^2(t) + \dot{P}_y^2(t)\right)^{1.5}} \tag{8}$$

In the path planning problem, Bezier curve is connected to form a smooth path planning for mobile robots.

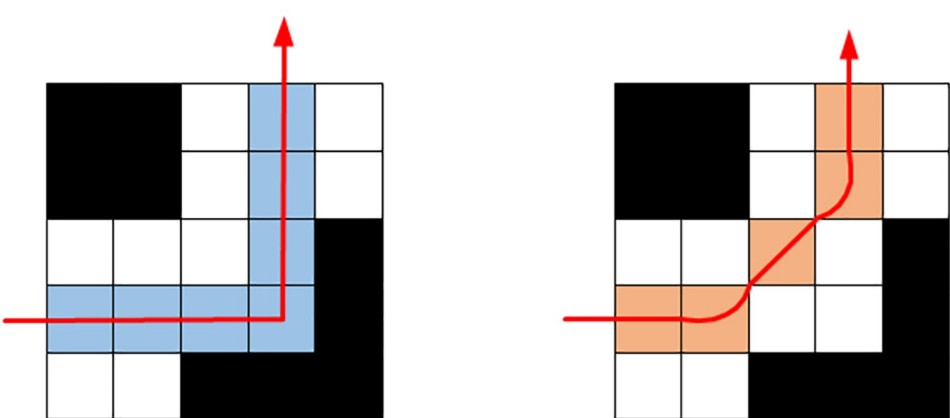

**Fig 4. Smoothing optimization strategy for a single right-angle turn.**

During the movement of the mobile robot, when the turning angle is greater than or equal to 90˚, the lengths of turning steps increase. This type of turn is generally divided into three steps: 1. decelerate and stop; 2. pivot toward the subsequent direction of travel; and 3. proceed forward. Both deceleration and acceleration of the mobile robot require engaging the steering gear, which greatly reduces the speed of the mobile robot.

To increase the practicability of the path reduce the number of such turns and improve the speed of the mobile robot, this section mainly optimizes the right-angle turns that are prone to occur during the path planning process. The basic idea is to decompose a 90˚ turn into multiple small-angle turns to improve the smoothness of the planned path. Since the traditional A* algorithm only considers the nodes in the four directions when searching the path, the turns are all right-angle turns. There are two cases corresponding to a single turn and a continuous turn:

When there is no obstacle on the inside of the corner to reduce the turning angle, the inflection point and its adjacent two nodes are replaced with adjacent points in the corner, and a 90˚ right-angle turn is decomposed into two 45˚ acute-angle turns, as shown in Fig 4.

1. When there is a continuous right-angle turn, the inflection point can be removed, and the two adjacent nodes of the inflection point can be directly connected to convert multiple right-angle turns into a small number of 45˚ acute-angle turns, as shown in Fig 5.

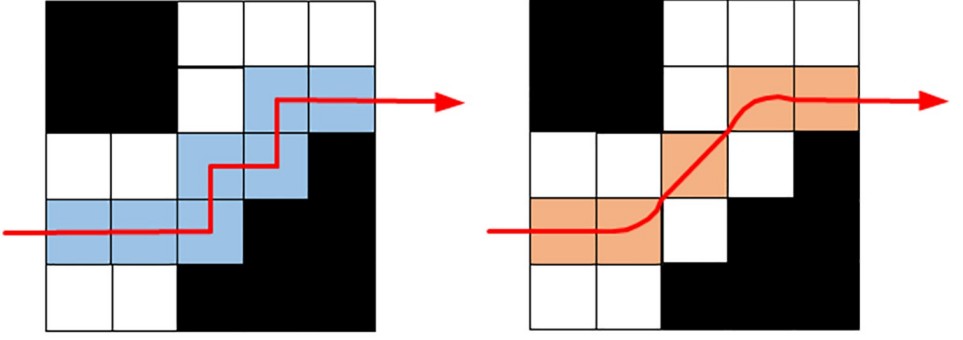

**Fig 5. Smoothing optimization strategy for continuous right-angle turns.**

2. The smoothing optimization of the right-angle turns improves the smoothness of the path and shortens the length of the path. It reduces the number of right-angle turns, which affects the turning efficiency and improves the overall efficiency of the robot.

## The EBS-A* algorithm pseudocode

The EBS-A* algorithm is implemented through the MATLAB programming language and software. The code is shown as Algorithm 1.

**Algorithm 1** The code of the EBS-A* Algorithm

```
Input: start node Start_i, end node End_i, environment map Map, esti-
       mated total costf(n)
Output: path PATH
1: Initialize Map,PATH_S, PATH_E
2: ExpansionDistance(1)
3: Create positive open list OPEN_LIST_1 and positive close list
   CLOSE_LIST_1
4: Create reverse open list OPEN_LIST_2 and reverse close list
   CLOSE_LIST_2
5: OPEN_LIST_1.add(Start_i), OPEN_LIST_2.add(End_i)
6: repeat
7:   node_s = OPEN_LIST_1.removeNext()
8:   PATH_S.add(node_s)
9:   search(node_s, OPEN_LIST_1, CLOSE_LIST_1)
10:  CLOSE_LIST_1.add(node_s)
11:  node_e = OPEN_LIST_2.removeNext()
12:  PATH_E.add(node_e)
13:  search(node_e, OPEN_LIST_2, CLOSE_LIST_2)
14:  CLOSE_LIST_2.add(node_s)
15: until node_s == End_i ‖ node_e == Start_i ‖ OPEN_LIST_1 == ∅ ‖
    OPEN_LIST_2 == ∅ ‖ GetNeighber(node_s) ∩ GetNeighber(node_e) ≠ ∅
16: PATH = PATH_S + PATH_E.reverse()
17: smooth(PATH)
18: return PATH
```

The execution process of the EBS-A* algorithm consists of four steps: 1. Performing expansion distance optimization on the A* algorithm. 2. Performing bidirectional search on the algorithm. 3. Generating an unsmooth path. 4. Performing smoothing process to generate a smooth path. The execution process of the proposed algorithm is shown in Fig 6.

## The EBS-A* algorithm time complexity analysis

In this section, combining the above bidirectional search and the smoothing optimization of right-angle turns, the pseudocode of the EBS-A* algorithm is provided as follows. The algorithm uses a double loop. The inner loop traverses adjacent nodes in four directions, and the

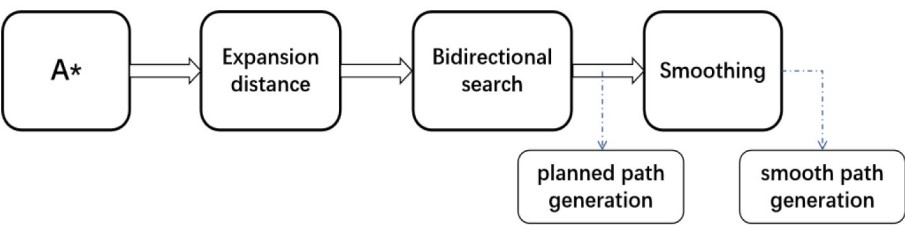

**Fig 6. The execution process of the EBS-A* algorithm.**

point with the least cost is marked and added to the open table. The outer loop traverses the
nodes of the open table until the queue traversal ends. There are several factors that exert obvi-
ous effects on the time complexity of various path planning algorithms, such as map scale,
starting node and target node location. In general, the time complexity of various algorithms is
given as follows:

$$T \in [O(min[m, n]), \ O(m * n)] \tag{9}$$

In formula 9, $m$ and $n$ are the length and width of the map, respectively. Since the bidirec-
tional search is a strategy to search in two directions, when the path is solvable, the path
formed by each search direction is smaller than the whole path, which reduces the number of
nodes added to the open table and decreases the loop frequency. Therefore, the time complex-
ity of the EBS-A* algorithm is less than or equal to half of that of the traditional A* algorithm.

## Simulation testing

### Experiment

In this section, we mainly simulate and test four algorithms, the traditional A* algorithm, the
A* algorithm with expansion distance, the bidirectional A* algorithm with expansion distance,
and the EBS-A* algorithm. The map scale is 50×50 in the simulation test, and the size of each
obstacle is 5×5 on the map. The location of obstacles is randomly generated on the map based
on the center point, but there are certain rules. The scale of the obstacle occupies a certain pro-
portion of the map scale, which is interpreted as the number of obstacle center points being
1% of the map scale. The four algorithms were tested, and the path planning results are shown
in Fig 7. The statistical results are shown in Table 1.

We use the 50×50 map to represent a moderately cluttered environment. In addition to test
on 50×50 maps, we conducted tests on 100×100 maps, 150×150 maps, and 200×200 maps. The
100×100 maps represent several obstacles and less cluttered environments, the 150×150 maps
represent a lot of obstacles and the highly cluttered environments, and the 200×200 map repre-
sents an environment with no obstacles. In different scale maps, algorithms are tested on a sin-
gle map and randomly generated maps and tested five times on randomized maps.

In Fig 7, the black blocks are randomly generated obstacles, the fluorescent blue blocks
around the obstacles are the expansion distances of the obstacles, the green and gray areas are
the nodes traversed by the forward and reverse searches, respectively, and the red line is the
final path 'generated by the EBS-A* algorithm.

Table 1 lists the running time of each algorithm required to plan the path, including the
time to complete all operations such as expansion and smoothing. This parameter indicates
the efficiency of the algorithm. The number of nodes and the total distance represent the
length of the planned path length. The number of right-angle turns and the maximum turning
angle correspond to the path smoothness and robustness, respectively. The number of expan-
sion nodes is the total number of nodes searched by the algorithm during path planning: this
parameter affects the efficiency of the algorithm. The number of critical nodes indicates the
number of nodes adjacent to obstacles in the path. The map scales are 50×50. The statistical
results on randomized maps are shown in Table 2, and all data are averages.

The experimental results show that the speed of the EBS-A* algorithm is improved by
approximately 328% compared with that of the conventional A* algorithm. The running time
of the bidirectional search algorithm is approximately 26.37% that of the unidirectional search
algorithm. The test results for other indicators are also in line with expectations. The random-
ized test results are consistent with the results in Table 2.

(a)

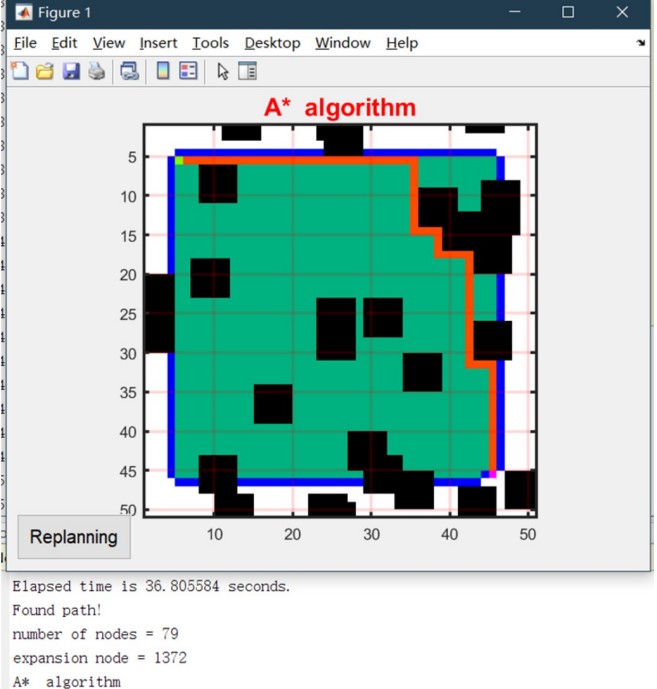

(b)

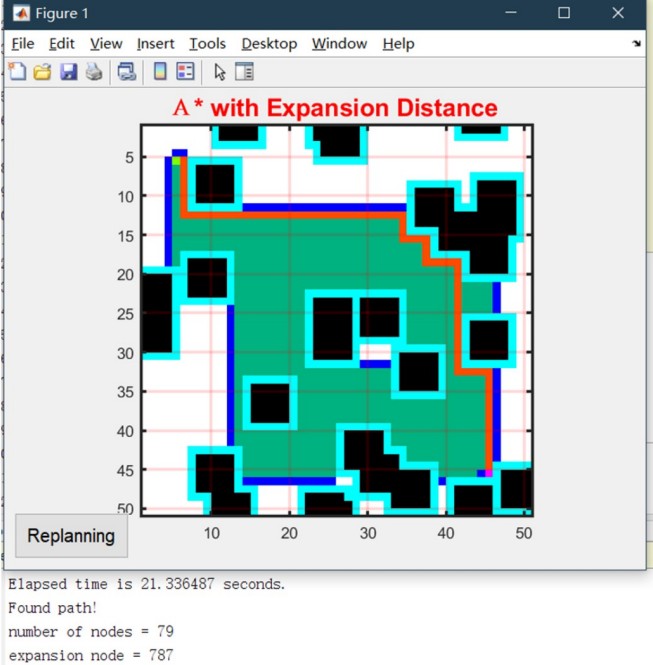

(c)

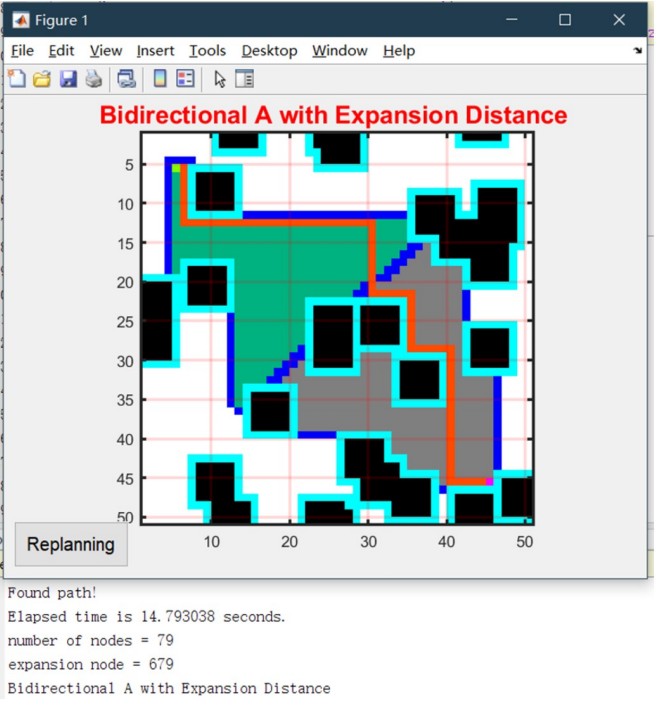

(d)

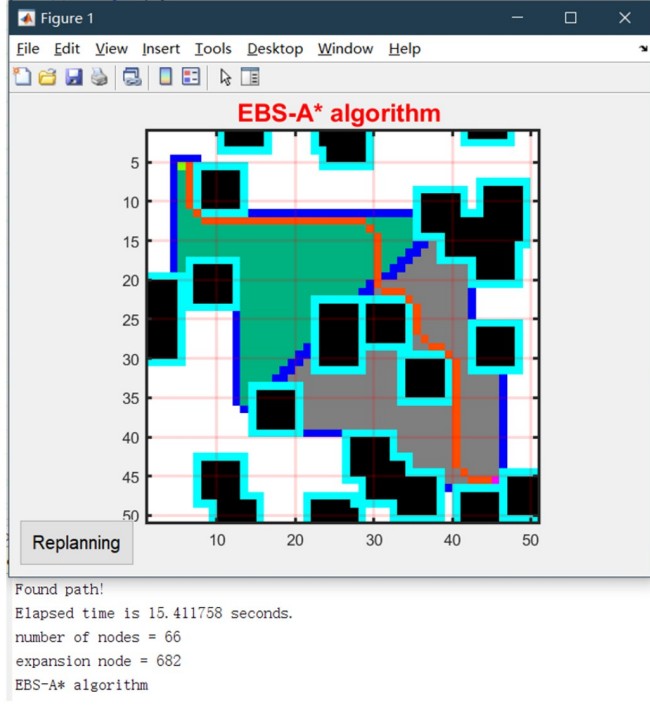

**Fig 7. Simulation results of four algorithms on a 50×50 map.** (a) A* algorithm. (b) A* with expansion distance. (c) Bidirectional A* with expansion distance. (d) EBS-A* algorithm.

**Table 1. Simulation results of four algorithms on a 50×50 map.**

| Indicators | A* algorithm | A* with Expansion Distance | Bidirectional A* with Expansion Distance | EBS-A* algorithm |
|---|---|---|---|---|
| Running time/s | 36.806(100%) | 21.782(59.18%) | 9.637(26.18%) | 9.747(26.48%) |
| Number of nodes | 79 | 79 | 79 | 66 |
| Number of right-angle turns | 7 | 9 | 8 | 0 |
| Max turning angle | 90˚ | 90˚ | 90˚ | 45˚ |
| Number of expansion nodes | 1372 | 787 | 679 | 682 |
| Number of critical nodes | 37 | 0 | 0 | 3 |

The path planning results of the 100×100 map are shown in Fig 8, the statistical results on single map are shown in Table 3, and the statistical results on randomized maps are shown in Table 4. The path planning results of the 150×150 map are shown in Fig 9, the statistical results on single map are shown in Table 5, and the statistical results on randomized maps are shown in Table 6. The path planning results of the 200×200 map are shown in Fig 10, the statistical results on single map are shown in Table 7, and the statistical results on randomized maps are shown in Table 8.

As shown in Table 3, the running time of the EBS-A* algorithm is 138.813s and the traditional A* algorithm is 302.467s on a 100×100 map. The efficiency of the EBS-A* algorithm is only 2.17 times that of the A* algorithm. As shown in Table 4, the efficiency of the EBS-A* algorithm is only 2.14 times that of the A* algorithm on randomized 100×100 maps. The statistical results in Tables 3 and 4 show that the test results of the algorithm on the fixed map and the random map are consistent. The efficiency of the algorithm is reliable.

But the efficiency of the EBS-A* algorithm is 4.7 times that of the A* algorithm on a 150×150 map. The reason is that there are only a few obstacles in the 100×100 map, but there are dense obstacles in the 150×150 map. In an environment with dense obstacles, the efficiency of the algorithm is higher. These statistical results have also been verified on 200×200 maps. the efficiency of the EBS-A* algorithm is 5.79 times that of the A* algorithm on a 200×200 map.

All the statistical results of the simulation test show that the efficiency of the EBS-A* algorithm is significantly improved compared with the traditional A* algorithm. These results verify the rationality of the algorithm design.

The geometric A* algorithm was proposed in [16] for AGV path planning. The algorithm was also optimized based on the traditional A* algorithm. The result is shown in Fig 11. To compare the performance of the EBS-A* and geometric A* algorithms, the rasterized map in [16] is reproduced, and the map scale is 100×100. The path planning result of the EBS-A* algorithm is shown in Fig 12.

**Table 2. Average simulation results of four algorithms on randomized 50×50 maps.**

| Indicators | A* algorithm | A* with Expansion Distance | Bidirectional A* with Expansion Distance | EBS-A* algorithm |
|---|---|---|---|---|
| Running time/s | 33.559(100%) | 21.386(63.73%) | 5.639(16.80%) | 7.835(23.35%) |
| Number of nodes | 79.4 | 79.4 | 79.4 | 67 |
| Number of right-angle turns | 6.4 | 6.2 | 7 | 0 |
| Max turning angle | 90˚ | 90˚ | 90˚ | 45˚ |
| Number of expansion nodes | 1,085 | 744.8 | 553 | 693.8 |
| Number of critical nodes | 36.2 | 0 | 0 | 3 |

(a)

(b)

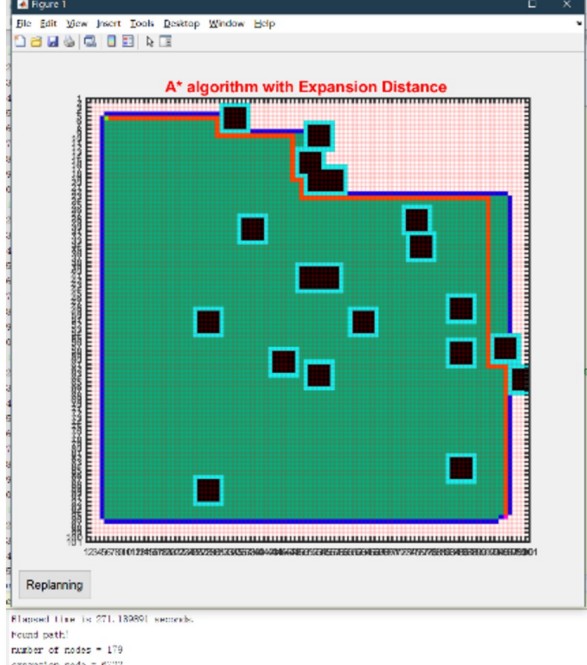

(c)

(d)

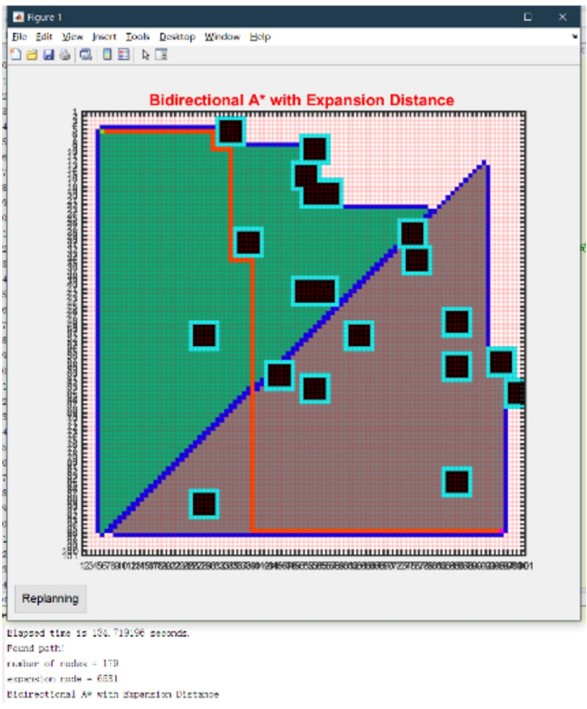

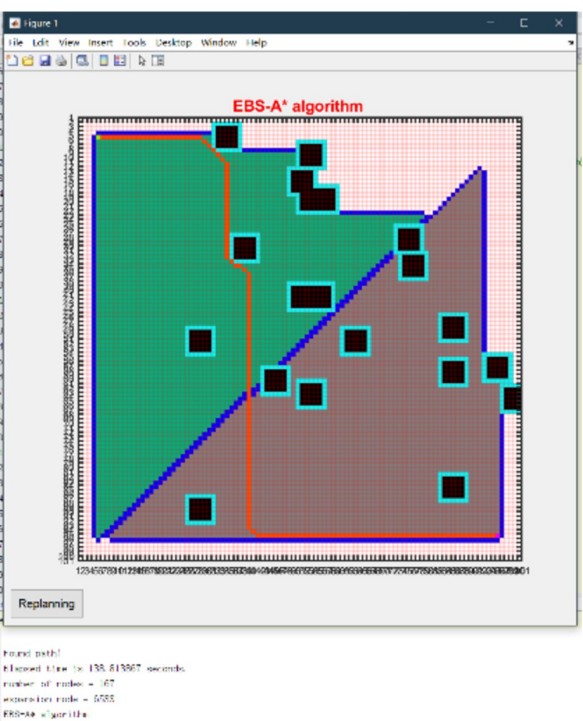

**Fig 8. Simulation results of four algorithms on a 100×100 map.** (a) A* algorithm. (b) A* with expansion distance. (c) Bidirectional A* with expansion distance. (d) EBS-A* algorithm.

**Table 3. Simulation results of four algorithms on a 100×100 map.**

| Indicators | A* algorithm | A* with Expansion Distance | Bidirectional A* with Expansion Distance | EBS-A* algorithm |
|---|---|---|---|---|
| Running time/s | 302.467 | 271.190 | 134.719 | 138.813 |
| Number of nodes | 179 | 179 | 179 | 167 |
| Number of right-angle turns | 7 | 9 | 6 | 0 |
| Max turning angle | 90˚ | 90˚ | 90˚ | 45˚ |
| Number of expansion nodes | 7,509 | 6722 | 6531 | 6533 |
| Number of critical nodes | 31 | 0 | 0 | 2 |

The experimental data in [16] are reproduced and the test results for the EBS-A* algorithm are summarized in Table 9.

As shown in Table 9, the experimental data in [16] are reproduced in the 2nd through 6th columns. It can be seen from the table that the path planning speed of the EBS-A* algorithm is 51.59 times that of the geometric A* algorithm, which is an enormous advantage. The running time of an algorithm is affected by many factors, such as computer performance, platform, and programming language. Therefore, the use of absolute time may be unacceptable.

In this research, we have tested the efficiency of the traditional A* algorithm and EBS-A* algorithm. In [16], authors also have tested the efficiency of the traditional A* algorithm and the geometric A* algorithm. We can choose the efficiency of the traditional A* algorithm as a benchmark to compare the efficiency of the EBS-A* algorithm and the geometric A* algorithm. As shown in Table 9, the running time of the A* algorithm is 316.334s and the running time of the geometric A* algorithm is 292.142s in [16]. The running time of the traditional A* algorithm is 36.806s and the running time of the EBS-A* algorithm is 9.747s. The histograms of EBS-A* and geometric A* in Fig 13 are the results of proportional calculations.

## Discussion

**Efficiency.** Here we discuss the efficiency of the algorithm. The experimental results show that the running time of the EBS-A* algorithm is approximately 26.48% that of the conventional A* algorithm, which means that the path planning efficiency is improved by 278%. The running time of a bidirectional search by the A* algorithm with an expansion distance is approximately 44.24% that of the unidirectional search by the A* algorithm with expansion distance, and the experimental result is consistent with the time complexity analysis introduced in the previous section.

In the comparison testing with Geometric A*, the traditional A* algorithm was also taken as the benchmark. The proportional comparison method was applied: that is, the running times of the A* algorithm and the EBS-A* algorithm were calculated, and their ratio was used to calculate the relative running time. The relative time was also used to compare the running

**Table 4. Average simulation results of four algorithms on randomized 100×100 maps.**

| Indicators | A* algorithm | A* with Expansion Distance | Bidirectional A* with Expansion Distance | EBS-A* algorithm |
|---|---|---|---|---|
| Running time/s | 269.335 | 268.331 | 128.775 | 125.677 |
| Number of nodes | 179 | 179 | 179 | 170.8 |
| Number of right-angle turns | 6 | 5.7 | 6 | 0 |
| Max turning angle | 90˚ | 90˚ | 90˚ | 45˚ |
| Number of expansion nodes | 7,709.4 | 7,302.8 | 6,817.4 | 6,599 |
| Number of critical nodes | 10 | 0 | 0 | 2 |

(a)

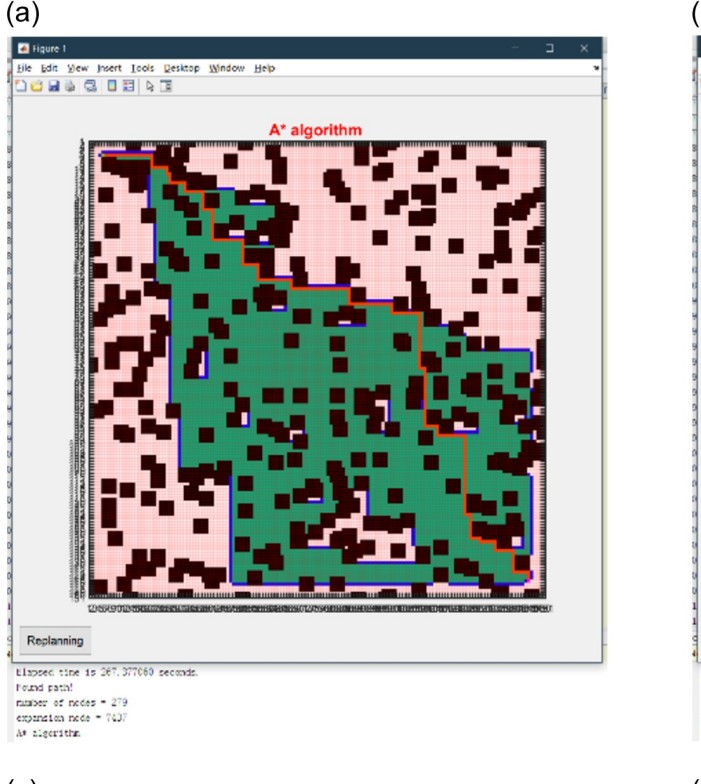

(b)

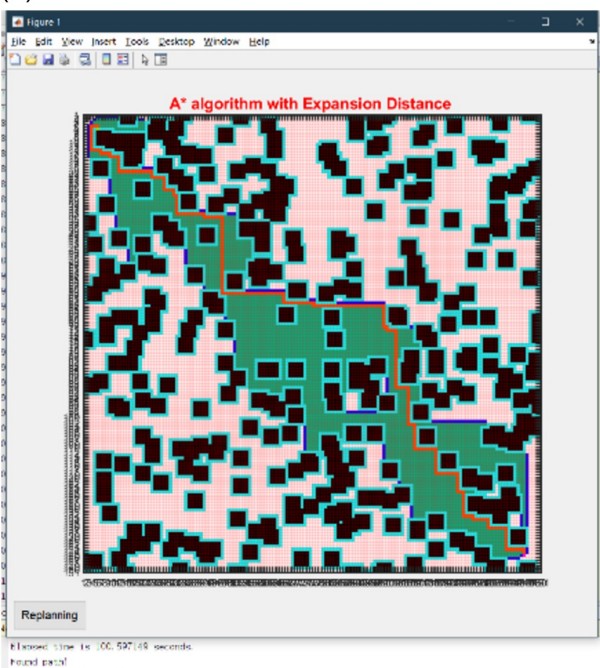

(c)

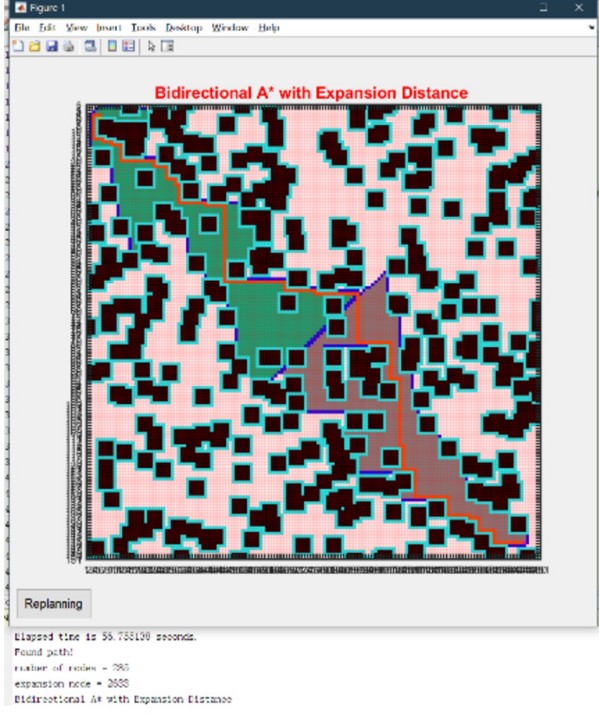

(d)

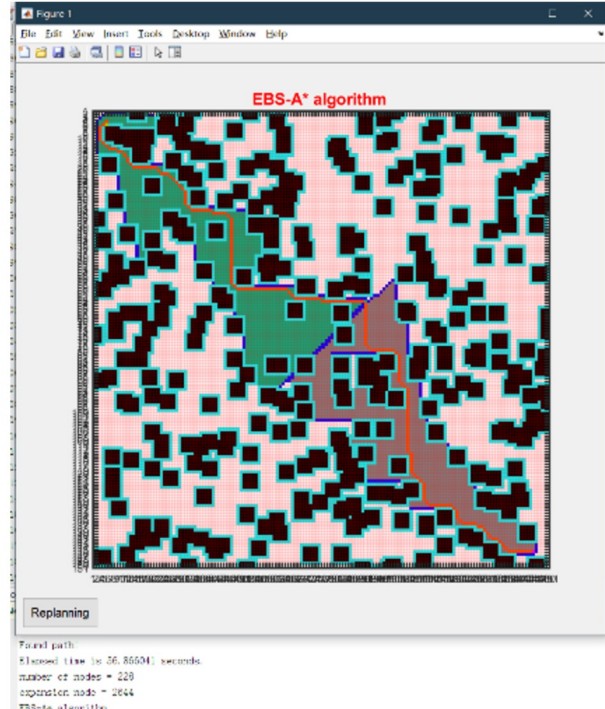

**Fig 9. Simulation results of four algorithms on a 150×150 map.** (a) A* algorithm. (b) A* with expansion distance. (c) Bidirectional A* with expansion distance. (d) EBS-A* algorithm.

**Table 5. Simulation results of four algorithms on a 150×150 map.**

| Indicators | A* algorithm | A* with Expansion Distance | Bidirectional A* with Expansion Distance | EBS-A* algorithm |
|---|---|---|---|---|
| Running time/s | 267.377 | 100.597 | 56.755 | 56.866 |
| Number of nodes | 279 | 285 | 285 | 228 |
| Number of right-angle turns | 39 | 47 | 42 | 0 |
| Max turning angle | 90° | 90° | 90° | 45° |
| Number of expansion nodes | 7,437 | 2,931 | 2,633 | 2,644 |
| Number of critical nodes | 192 | 0 | 0 | 11 |

time of the geometric A* algorithm. The relative running time conversion results are shown in Fig 13. After the calculation, the speed of the EBS-A* algorithm is 3.52 times that of the geometric A* algorithm, which still provides a large advantage. This is one of the original intentions that guided the design of the EBS-A* algorithm.

The running time of the EBS-A* algorithm is longer than that of the bidirectional search with an expansion distance because the smoothing process introduced here first traverses the planned path and smooths it, if there is a right-angle turn. Therefore, the EBS-A* algorithm sacrifices the running time at a cost that is justified because the path robustness is effectively enhanced. Through multiple simulation verifications on different map scales, the EBS-A* algorithm has shown excellent performance and consistency.

**Robustness.** Table 1 shows that the number of path nodes is reduced by 16.46%, while all right-angle turns are smoothed, and the maximum turning angle is 45°. As shown in Table 2, with the conventional A* algorithm, the number of critical nodes is 37 in the original map and is reduced to 0 after employing the expansion distance. The strategy effectively increases the path robustness. Moreover, with the EBS-A* algorithm, the number of critical nodes is increased by 3 because the specific path environment needs to borrow critical points for path smoothing. This situation does not cause a significant decrease in the path robustness, which only occurs in a few cases, and the borrowed critical points are only at the corners of the obstacles, which have been smoothed. In combination, with the EBS-A* algorithm, the number of critical points is reduced by 91.89%.

Compared with the other algorithms, the EBS-A* algorithm proposes the least number of turns, which is 18, and the path smoothness is better. Since the expansion distance is employed in the EBS-A* algorithm, critical nodes are used to build barriers, thereby effectively avoiding collisions and guaranteeing path robustness. The critical node of the optimized path is 0, and other algorithms have not adopted corresponding protection strategies in performing this test. As seen from Fig 12, the robustness of the path is guaranteed in the map with dense obstacles due to protection by the expansion distance. However, in Fig 11, multiple sections of the geometric A* algorithm path are close to the obstacles, and the robustness of the path is greatly threatened. Therefore, the robustness of the EBS-A* algorithm has a greater advantage than

**Table 6. Average simulation results of four algorithms on randomized 150×150 maps.**

| Indicators | A* algorithm | A* with Expansion Distance | Bidirectional A* with Expansion Distance | EBS-A* algorithm |
|---|---|---|---|---|
| Running time/s | 264.789 | 116.379 | 45.381 | 57.032 |
| Number of nodes | 279.4 | 284.2 | 283.8 | 233.4 |
| Number of right-angle turns | 29 | 26.2 | 28 | 0 |
| Max turning angle | 90° | 90° | 90° | 45° |
| Number of expansion nodes | 8,864.4 | 3,713.2 | 2,713 | 3,325.6 |
| Number of critical nodes | 201 | 0 | 0 | 12 |

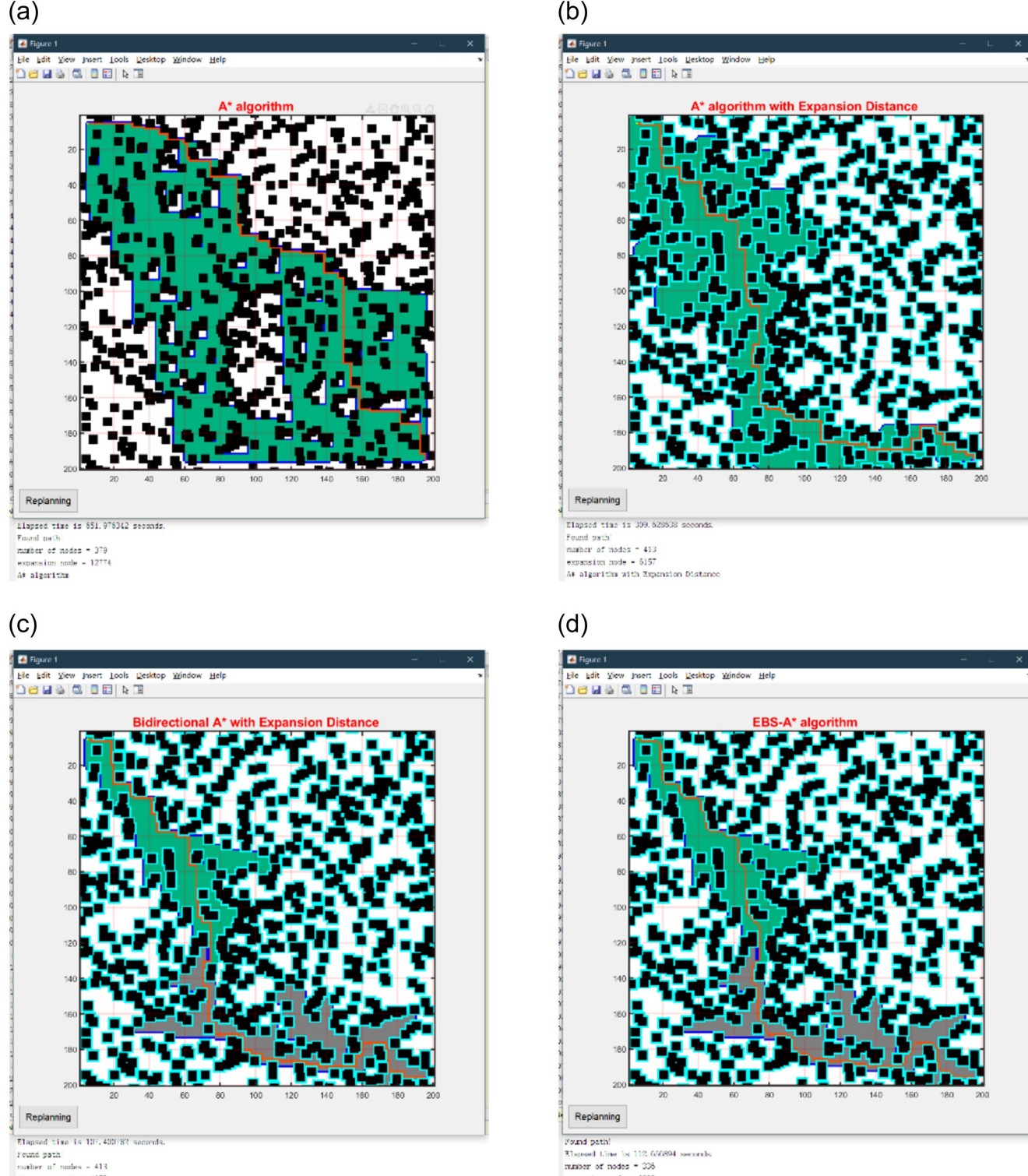

**Fig 10. Simulation results of four algorithms on a 200×200 map.** (a) A* algorithm. (b) A* with expansion distance. (c) Bidirectional A* with expansion distance. (d) EBS-A* algorithm.

**Table 7. Simulation results of four algorithms on a 200×200 map.**

| Indicators | A* algorithm | A* with Expansion Distance | Bidirectional A* with Expansion Distance | EBS-A* algorithm |
|---|---|---|---|---|
| Running time/s | 651.976 | 309.629 | 107.401 | 112.656 |
| Number of nodes | 379 | 413 | 413 | 336 |
| Number of right-angle turns | 45 | 61 | 62 | 0 |
| Max turning angle | 90° | 90° | 90° | 45° |
| Number of expansion nodes | 12,774 | 6,157 | 4,273 | 4,288 |
| Number of critical nodes | 253 | 0 | 0 | 13 |

**Table 8. Average simulation results of four algorithms on randomized 200×200 maps.**

| Indicators | A* algorithm | A* with Expansion Distance | Bidirectional A* with Expansion Distance | EBS-A* algorithm |
|---|---|---|---|---|
| Running time/s | 674.438 | 378.412 | 116.365 | 119.974 |
| Number of nodes | 380.6 | 443 | 414.6 | 330.6 |
| Number of right-angle turns | 43 | 59.2 | 60 | 0 |
| Max turning angle | 90° | 90° | 90° | 45° |
| Number of expansion nodes | 11,809.8 | 6,598.4 | 5,238.2 | 4,426.8 |
| Number of critical nodes | 275 | 0 | 0 | 16 |

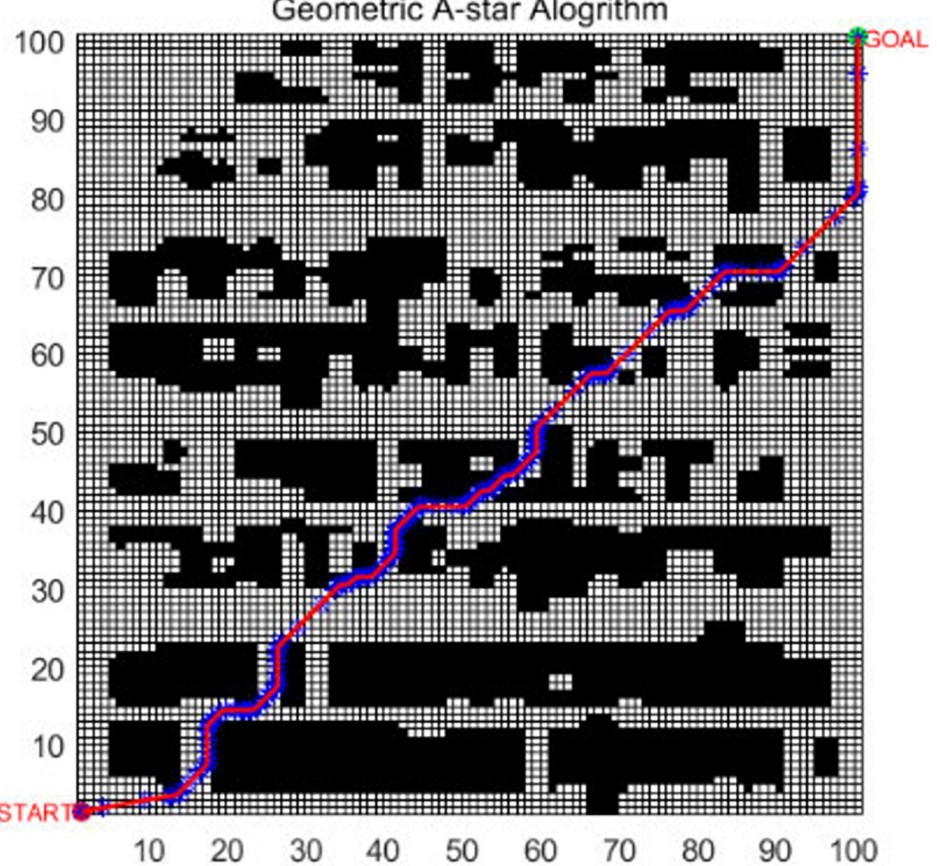

**Fig 11. The path planning of the geometric A*.**

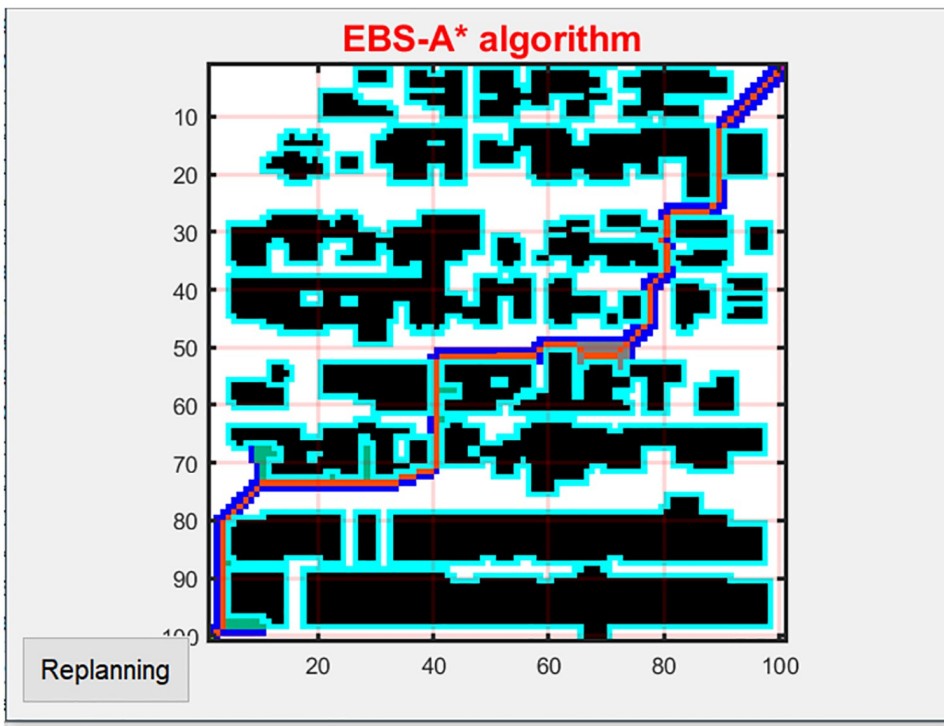

**Fig 12. The path planning of the EBS-A***.

the other algorithms, including the geometric A* algorithm. This is another goal pursued by the developers of the EBS-A* algorithm. Since the EBS-A* algorithm did not pursue the shortest path as the goal at the beginning of the design, it is not advantageous in terms of the number of nodes and the total distance. The design idea of the algorithm is to first ensure the robustness of the path; otherwise, the cost of collisions is far greater than the performance advantages of other indicators. Therefore, the EBS-A* algorithm gains efficiency and robustness at the cost of path length.

**Table 9. Algorithm performance comparison.**

| Indicators | A* | BFS | Dijkstra | DFS | Geometric A* | EBS-A* |
|---|---|---|---|---|---|---|
| Running time/s | 316.334 | 322.962 | 316.334 | 394.83 | 295.142 | 9.747 |
| Number of nodes | 131 | 131 | 131 | 198 | 109 | 161 |
| Total distance | 158.167 | 161.481 | 158.167 | 197.415 | 147.571 | 182.426 |
| Number of turns | 36 | 33 | 27 | / | 27 | 18 |
| Max turning angle | 45° | 135° | 45° | 90° | 45° | 45° |
| Number of expansion nodes | 2246 | 5936 | 6047 | 198 | 109 | 219 |
| Number of critical nodes | / | / | / | / | / | 0 |

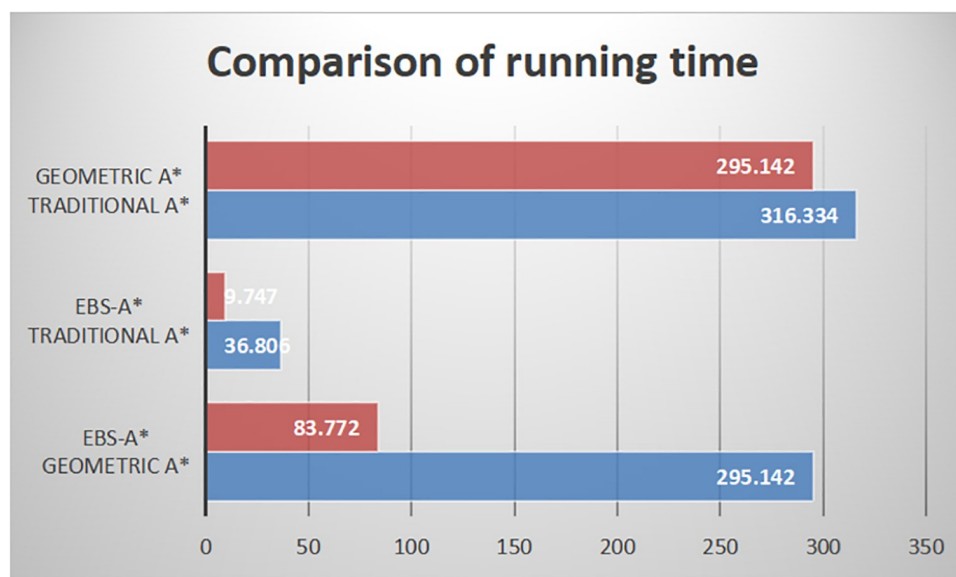

**Fig 13. Comparison graph of algorithm running time.**

The analysis of the test results from a less cluttered to highly cluttered environment shows that in a more cluttered environment, the advantages of the EBS-A* algorithm are more obvious than those of the traditional A* algorithm in terms of the robustness index, and the performance of the algorithm is collision-free. The difference between the EBS-A* algorithm and the traditional A* algorithm is that the efficiency of the algorithm is improved by expanding the distance and bidirectional search, and the robustness of the algorithm is enhanced by expanding the distance and smoothing. An environment without obstacles is a collision-free environment, if the robot will not collide during travel, the robustness of the algorithm will not be affected, and the advantage of the EBS-A* algorithm will no longer be obvious. In a barrier-free environment, there is no fundamental difference between the paths planned by the two algorithms, and the efficiency of bidirectional search will be twice that of single search, and the algorithm will strictly follow this rule.

## Real-world case

The excellent performance of the EBS-A* algorithm has been verified through simulation testing. To verify the application potential of the algorithm in real scenarios, we select a mobile robot as a real-world case to verify the effectiveness of the algorithm.

The hardware platform used in this experiment is the FS-AIROBOTB intelligent robot. The hardware composition is shown in Fig 14, and the description of each component is shown in Table 10.

To test the effectiveness of the EBS-A* algorithm, it is transplanted to the FS-AIROBOTB mobile robot hardware platform in this paper. Given the open-source nature of the ROS and the fact that the ROS contains the navigation function package "nav_ROS" for path planning, the original algorithm of the function package is rewritten to realize the transplant of the algorithm. This method can achieve the advantages of rapid development and tighter integration with other components of the algorithm.

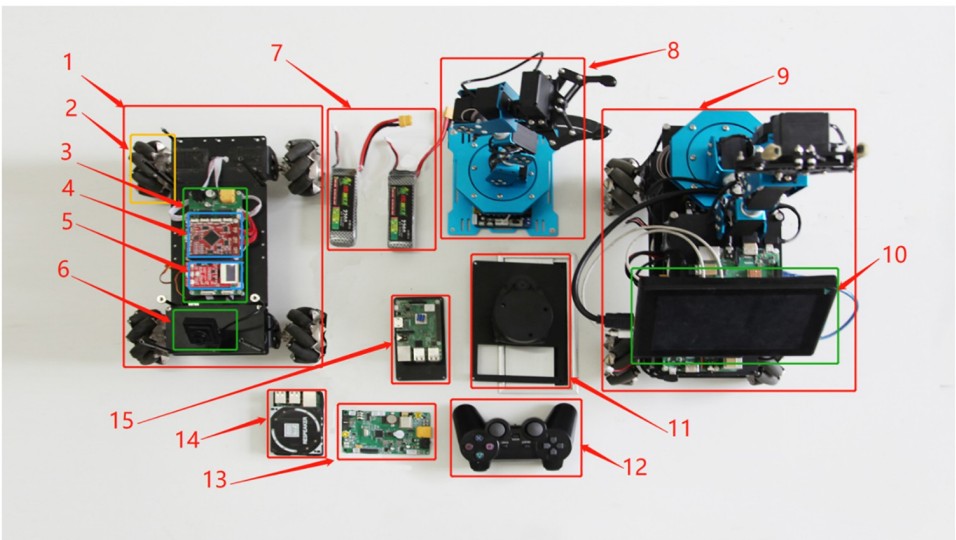

**Fig 14. FS-AIROBOTB component diagram.**

The start node and the end node are set in the real world. The robot uses the EBS-A* algorithm to plan a collision-free path to the goal position. This experiment process consists of three steps:

1. EBS-A* algorithm transplantation;

2. simultaneous localization and mapping (SLAM) test;

3. a robot autonomous navigation test.

Algorithm transplantation accomplishes writing EBS-A* algorithm into the ROS. The SLAM test built a test map to use the radar on the robot in the real world. The autonomous navigation test verifies the effectiveness of the algorithm in the real world. This section introduces the implementation of autonomous robot navigation experiments through algorithm transplantation and map construction, to verify the effectiveness of EBS-A* algorithm in the real world. The actual test environment and the map constructed by the SLAM are shown in Fig 15.

As shown in Fig 16, the left picture is the rviz interface of the ROS navigation, and the right picture is the extracted path. The result shows that the EBS-A* algorithm is feasible for the ROS of the robot and that the planned path is smooth.

**Table 10. FS-AIROBOTB component serial number comparison table.**

| Serial number | Part name | Serial number | Part name |
|---|---|---|---|
| 1 | ROS omnidirectional vehicle chassis | 9 | FS_AIROBOTB |
| 2 | Mecanum wheel | 10 | 7 inch HDMI display |
| 3 | Omnidirectional vehicle drive module | 11 | 360 degree lidar |
| 4 | Cortex-M4 chassis core control board | 12 | Wireless bluetooth remote control handle |
| 5 | FS_Explore sensor board | 13 | Cortex-M3 robotic arm control board |
| 6 | 1080P industrial module camera | 14 | 4 array microphones |
| 7 | Grep 3S/25C/1300mA power lithium battery | 15 | 3B+ Raspberry Pi |
| 8 | Six degrees of freedom robotic arm | | |

(a)

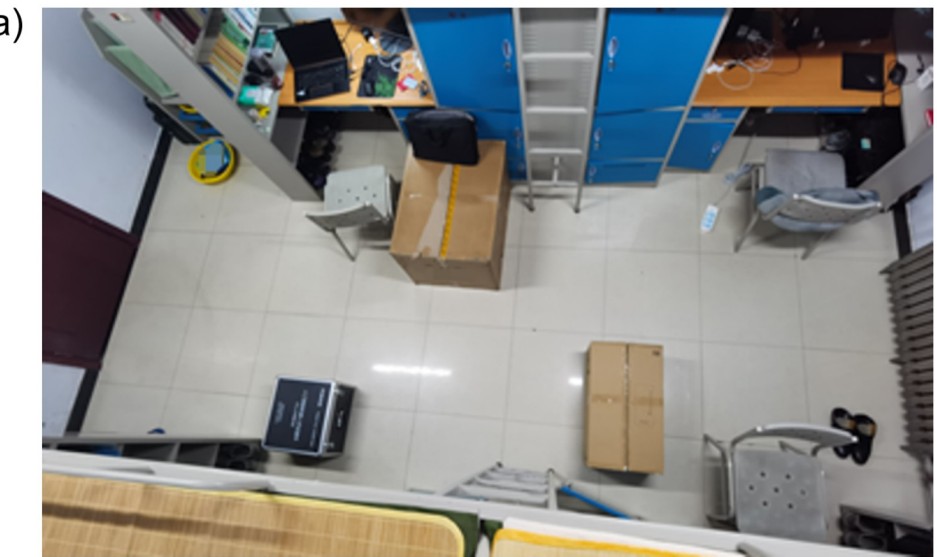

(b)

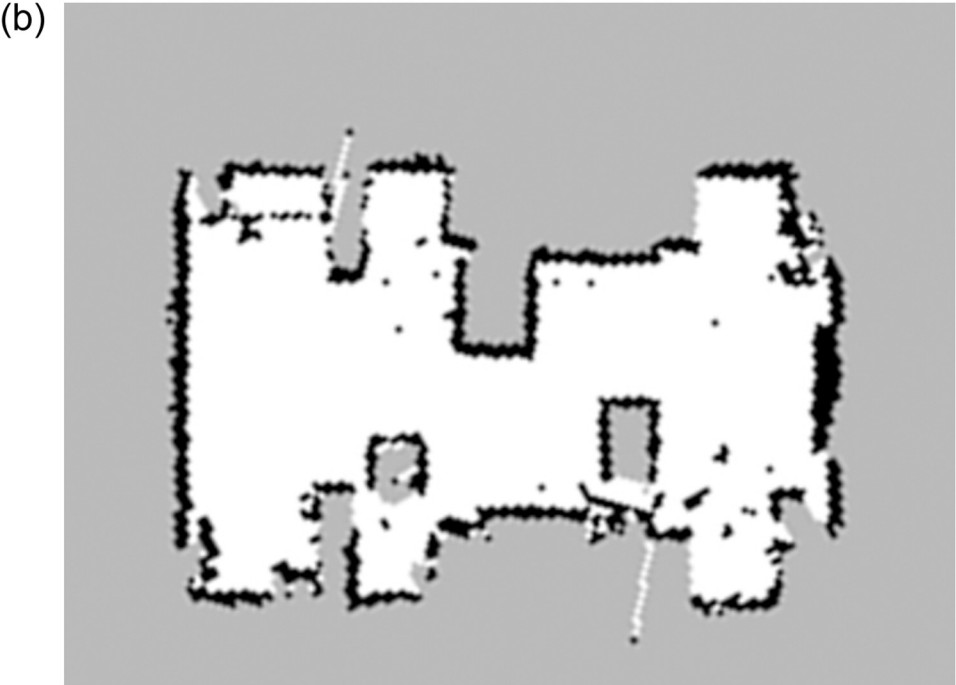

**Fig 15. Actual test environment and the map constructed by SLAM.**

In this experiment, we set an application environment for EBS-A* algorithm and carried out the autonomous navigation test. The EBS-A* algorithm was written into a real mobile robot. The experimental result shows that the robot can independently plan a reliable and smooth path and complete the autonomous navigation from the starting node to the goal node. This experiment verifies that the EBS-A* algorithm can be applied to mobile robots and has the potential to be applied to industrial scenarios. To enable other researchers to reproduce

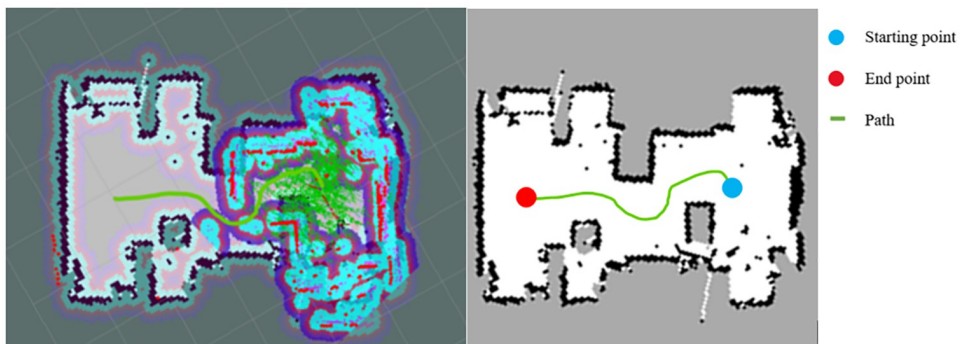

**Fig 16. The rviz interface of path planning and extracted path.**

our experimental process, we share the source and instructions of this paper through the following Github link: https://github.com/wanghw1003/EBAStar.

## Conclusions

In this paper, an improved path planning algorithm based on the conventional $A^*$ algorithm for mobile robots, named the EBS-$A^*$ algorithm, is proposed. This algorithm employed the three strategies of expansion distance, bidirectional search, and smoothing to overcome the limitations of the conventional $A^*$ algorithm in terms of path robustness and path planning efficiency. First, the expansion distance is set for the obstacles in the map to ensure that there is a fault-tolerant distance between the path and the obstacles. Second, a bidirectional search is added to the conventional $A^*$ algorithm to improve the speed of path planning. Third, the right-angle turns are smoothed and optimized in the path. Experimental results show that the EBS-$A^*$ algorithm improves the path planning efficiency by 278% and reduces the number of critical nodes by 91.89% and the number of right-angle turns by 100%.

## Author Contributions

**Conceptualization:** Wei Liu.

**Formal analysis:** Jing Jing.

**Funding acquisition:** Tieming Liu.

**Methodology:** Huanwei Wang.

**Resources:** Yisen Wang.

**Validation:** Shangjie Lou.

**Writing – original draft:** Huanwei Wang.

**Writing – review & editing:** Jing Jing, Wei Liu.

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
