## [Decision Letter · Decision Letter 0]

8 Oct 2021

PONE-D-21-28660The EBS-A* algorithm: an improved A* algorithm for path planningPLOS ONE

Dear Dr. Wang,

Thank you for submitting your manuscript to PLOS ONE. After careful consideration, we feel that it has merit but does not fully meet PLOS ONE’s publication criteria as it currently stands. Therefore, we invite you to submit a revised version of the manuscript that addresses the points raised during the review process.

We look forward to receiving your revised manuscript.

Kind regards,

Yogendra Arya

Academic Editor

PLOS ONE

Journal Requirements:

Reviewers' comments:

Reviewer's Responses to Questions

**Comments to the Author**

1. Is the manuscript technically sound, and do the data support the conclusions?

Reviewer #1: Yes

Reviewer #2: Yes

2. Has the statistical analysis been performed appropriately and rigorously? 

Reviewer #1: Yes

Reviewer #2: No

3. Have the authors made all data underlying the findings in their manuscript fully available?

Reviewer #1: Yes

Reviewer #2: Yes

4. Is the manuscript presented in an intelligible fashion and written in standard English?

Reviewer #1: Yes

Reviewer #2: No

5. Review Comments to the Author

Reviewer #1: In section 2 (Related work), when explaining the related literature and the proposed A*-based algorithms' difficulties, please, mention how your proposed method handles those difficulties and contributes to the corresponding literature.

How the expansion distance is automatically decided for different environments? Is it only one extra grid cell from the sides of the obstacle or it can include multiple layers of grid cells on the sides of the obstacle? how does the algorithm decide to choose the width of the expansion distance in accordance with the environmental scale?

Can Fig.2 be modified to see how the proposed bidirectional search handles the situation when there is an obstacle inside the grids?

On page 16, " and the number is 1% of the map scale", is not clear.

I would like to see 4 sets of simulation tests similar to Fig. 5 (including it) with different map scales (100x100, 150x150x, 200x200x) and a different number of obstacles (demonstrating less cluttered to highly cluttered environments) to confirm the performance of the proposed method.

Reviewer #2: Please consider the following comments to improve your manuscript before it can be considered for publication:

1. For related work section, you should discuss the algorithms based on their classification either classical or meta-heuristic or other. Then summarize your findings.

2. There is no proper justification on why A* is improved rather than Artificial Potential Field or D* where have better performance compare to A* in some literatures.

3. The pseudocode format is incorrect. Please correct them.

4. Why Expansion distance and Bidirectional search optimization are considered to be hybridized with A*? Please justify them with proper support.

5. I would like to suggest at least three scenarios for simulation test (no obstacle, several obstacles, and a lot of obstacles).

6. Comparison graphs to compare the performance of the algorithms proposed against the existing one.

7. There is no significant analysis done by the authors. Please consider them.

8. The effectiveness test only utilize EBS-A* algorithm. Why not other existing algorithm as well?

9. Please proof read your manuscript before your resubmission.

6. PLOS authors have the option to publish the peer review history of their article (what does this mean?). If published, this will include your full peer review and any attached files.

Reviewer #1: **Yes: **Hossein B. Jond

Reviewer #2: No

---

## [Author Response · Author response to Decision Letter 0]

4 Nov 2021

Response to Reviewer 1 Comments

Point 1: In section 2 (Related work), when explaining the related literature and the proposed A*-based algorithms' difficulties, please, mention how your proposed method handles those difficulties and contributes to the corresponding literature.

Response 1:

we have completely rewritten the Related work section and reorganized and analyzed the references, it mainly focuses on the efficiency and robustness of the algorithm. Firstly, we analyzed several improved A* algorithms and these variants in efficiency and summarized the existing shortcomings. Secondly, we analyze several literatures to enhance the reliability of the algorithm and some methods, such as smoothing, and summarize the shortcomings of these improved algorithms. We found that the performance of the existing A* improved algorithm is not balanced, and lacks the A* improved algorithm with high efficiency and strong robustness. Based on the analysis and summary of the references, we propose a combined improvement strategy of the A* algorithm.

Point 2: How the expansion distance is automatically decided for different environments? Is it only one extra grid cell from the sides of the obstacle or it can include multiple layers of grid cells on the sides of the obstacle? how does the algorithm decide to choose the width of the expansion distance in accordance with the environmental scale?

Response 2:

It is not only one extra grid cell from the sides of the obstacle, multiple layers of grid cells can be included on the sides of the obstacle. Regarding the question of how many layers of grid cells extend around the obstacle, this appropriate distance is related to the robot size, the map scale, and the traveling speed of the robot. How to automatically decide the expansion distance in different environments? We simplified and equivalent to a model, and made a simplified model of the robot and obstacles. Based on these constraints, we constructed the mathematical relationship between the expansion distance and the cruising speed of the robot. Based on this knowledge, the expansion distance can be decided automatically and reasonably.

Point 3: Can Fig.2 be modified to see how the proposed bidirectional search handles the situation when there is an obstacle inside the grids?

Response 3:

We have added two graphs, respectively, when there is one obstacle and multiple obstacles inside the grids. Through these graphs, we can clearly understand how the bidirectional search handles the situations when encountering obstacles.

Point 4: On page 16, " and the number is 1% of the map scale", is not clear.

Response 4:

We have revised the unclear words.

After revised: The location of obstacles is randomly generated on the map based on the center point, but there are certain rules. The scale of the obstacle occupies a certain proportion of the map scale, which is interpreted as the number of obstacle center points is 1% of the map scale.

Point 5: I would like to see 4 sets of simulation tests similar to Fig. 5 (including it) with different map scales (100x100, 150x150x, 200x200x) and a different number of obstacles (demonstrating less cluttered to highly cluttered environments) to confirm the performance of the proposed method.

Response 5:

We have added three groups of simulation tests on different map scales (100x100, 150x150x, and 200x200), including the tests of fixed maps and the multiple tests of randomized maps. Set up obstacles of different densities on maps of different scales to represent from less cluttered to highly cluttered environments. We have analyzed the experimental results, and the algorithm has shown results in line with expectations on different scale maps.

 

Response to Reviewer 2 Comments

Point 1: For related work section, you should discuss the algorithms based on their classification either classical or meta-heuristic or other. Then summarize your findings.

Response 1: 

we have completely rewritten the Related work section and rediscussed the classification of path planning algorithms. We reorganized and analyzed the references, focusing on the efficiency and robustness of the algorithm, and summarized the existing shortcomings of improved A* algorithms and these variants. We found that the performance of the existing improved A* algorithms is not balanced, and lacks the improved A* algorithm with high efficiency and strong robustness. Based on the analysis and summary of the literatures, we propose a combined improvement strategy of the A* algorithm.

Point 2: There is no proper justification on why A* is improved rather than Artificial Potential Field or D* where have better performance compare to A* in some literatures.

Response 2: 

The application scenario of this research is path planning in static planning (or global path planning), and the A* algorithm is one of the most well-known and most widely used algorithms in static path planning. The Artificial Potential Field and D* are path planning algorithms used in dynamic planning scenarios. The revised content of the manuscript is as follows:

Path planning algorithms are also divided into global path planning and local path planning based upon available environmental knowledge. Global path planning seeks an optimal path given largely complete environmental information and is best performed when the environment is static and perfectly known to the robot. Therefore, global path planning is also called static path planning. By contrast, local path planning is most typically performed in unknown or dynamic environments, local path planning is also called dynamic path planning.

For application scenarios such as warehousing and logistics, path planning in a static environment assumes that the robot perceives the environment and uses local path planning algorithms when the environmental information is not fully grasped. A* is used for shortest path evaluation based on the information regarding the obstacles present in the static environment[10], and the shortest path evaluation for the known static environment is a two-level problem, which comprises a selection of feasible node pairs and shortest path evaluation based on the obtained feasible node pairs[11]. Both of the above-mentioned criteria are not available in a dynamic environment, which makes the algorithm inefficient and impractical in dynamic environments. In this case, the dynamic path planning algorithm is not suitable for use in a static environment. Classical algorithms include D* algorithm, Artificial Potential Field algorithm, and etc. A* algorithm is chosen because it represents the foundational algorithms used within contemporary real-time path planning solutions in a static environment. Novel research builds on the algorithm to find additional performance and efficiency.

Point 3: The pseudocode format is incorrect. Please correct them.

Response3: 

We have revised the pseudocode format to keep it in the correct format.

Point 4: Why Expansion distance and Bidirectional search optimization are considered to be hybridized with A*? Please justify them with proper support.

Response 4: 

We have added the theory of expansion distance and Bidirectional search in this manuscript to support the rationality of the proposed optimization strategy. Since A* algorithm is a path planning algorithm based on graph search and evolved from BFS, therefore, it is reasonable to apply two optimization strategies based on graph node operation to improve the A* algorithm.

Point 5: I would like to suggest at least three scenarios for simulation test (no obstacle, several obstacles, and a lot of obstacles).

Response 5: 

We have added three groups of experiments on different map scales (100*100, 150*150, 200*200). Like the experiments in the manuscript, we performed the tests in fixed maps and randomized maps. Set up obstacles of different densities on maps of different scales to represent from less cluttered to highly cluttered environments. However, we did not add the tests in no obstacle scenarios, our considerations are as follows:

The difference between the EBS-A* algorithm and the traditional A* algorithm is that the efficiency of the algorithm is improved by expanding the distance and bidirectional search, and the robustness of the algorithm is enhanced by expanding the distance and smoothing. An environment without obstacles is a collision-free environment, if the robot will not collide during the travel, the robustness of the algorithm will not be affected, and the advantage of the EBS-A* algorithm will no longer be obvious. In a barrier-free environment, there is no fundamental difference between the paths planned by the two algorithms, and the efficiency of bidirectional search will be twice that of single search, and the algorithm will strictly follow this rule.

Point 6: Comparison graphs to compare the performance of the algorithms proposed against the existing one.

Response 6: 

We have added the comparison graph to compare the performance of the algorithms proposed against the existing one. The graph is as follow:

Point 7: There is no significant analysis done by the authors. Please consider them.

Response 7: 

We adjusted the chapter structure, added the discussion section, moved the part that involved analysis and discussion in the original simulation test section to discussion section. We divide the discussion into two sections, namely efficiency and robustness, to discuss separately. The fixed maps, randomized maps, and comparative experiments were discussed and analyzed in turn. It also gives a general analysis of the algorithm in the environment of different density obstacles.

Point8: The effectiveness test only utilize EBS-A* algorithm. Why not other existing algorithm as well?

Response 8: 

The effectiveness test is to verify the effectiveness and feasibility of the proposed algorithm in mobile robots and real physical scenarios and to verify whether the algorithm has the potential for industrial applications. Simulation test is only a way to verify the performance of the algorithm, and the real hardware platform test is to more comprehensively verify the application value of the algorithm and the potential of industrial applications. After the reviewers’ comment and in-depth consideration, the section called the Effectiveness test is indeed unreasonable, so it is called an application case. We have modified this section as an application case and made slight changes to the content.

Point 9: Please proof read your manuscript before your resubmission.

Response 9: 

We adopted the reviewer's comments and carefully proofread the full text of the manuscript based on extensive revisions.

---

## [Decision Letter · Decision Letter 1]

7 Dec 2021

PONE-D-21-28660R1The EBS-A* algorithm: an improved A* algorithm for path planningPLOS ONE

Dear Dr. Wang,

Thank you for submitting your manuscript to PLOS ONE. After careful consideration, we feel that it has merit but does not fully meet PLOS ONE’s publication criteria as it currently stands. Therefore, we invite you to submit a revised version of the manuscript that addresses the points raised during the review process.

We look forward to receiving your revised manuscript.

Kind regards,

Yogendra Arya

Academic Editor

PLOS ONE

Additional Editor Comments (if provided):

Still the work is not publishable. The manuscript requires further major revision.

Reviewers' comments:

Reviewer's Responses to Questions

**Comments to the Author**

1. If the authors have adequately addressed your comments raised in a previous round of review and you feel that this manuscript is now acceptable for publication, you may indicate that here to bypass the “Comments to the Author” section, enter your conflict of interest statement in the “Confidential to Editor” section, and submit your "Accept" recommendation.

Reviewer #1: All comments have been addressed

Reviewer #2: All comments have been addressed

2. Is the manuscript technically sound, and do the data support the conclusions?

Reviewer #1: Partly

Reviewer #2: Yes

3. Has the statistical analysis been performed appropriately and rigorously? 

Reviewer #1: Yes

Reviewer #2: No

4. Have the authors made all data underlying the findings in their manuscript fully available?

Reviewer #1: Yes

Reviewer #2: Yes

5. Is the manuscript presented in an intelligible fashion and written in standard English?

Reviewer #1: Yes

Reviewer #2: Yes

6. Review Comments to the Author

Reviewer #1: My previous comments have been addressed. However, I still have a number of comments which are given below.

1. The English of the manuscript still requires improvement.

2. Line 48, and line 53 need correction.

3. Line 60-61, what do those abbreviations stand for?

4. Very long sentence from line 71 to 75.

5. the paragraph starting from line 128 and its intentions are unclear.

6. the paragraph starting from line 152 is missing a strong motivation for this study.

6. line 156 "The A* algorithm was first proposed and described in detail in [13]," should be double fact-checked. A* has been proposed firstly half a century ago.

7. Line 175 needs correction.

8. In formula (2), the sub index of E(V_i) should be corrected, i.e., should be without underscore. The * (if denotes multiplication) is redundant and should be removed. Throughout the paper, often the text format of parameters in formulas and their descriptions in the text do not match.

9. The assumption in line 216 does not make sense. Assuming that the side length of the square is equal to the radius of the robot is a very poor assumption since the obstacles can vary in size greatly. Assuming all obstacles are identical in size with the robot is not realistic.

10. In formula (2), Vi is introduced as the current cruising speed of the robot and Vr is the cruising speed of the robot. What is the difference between these two speeds? It looks likes they refer to an identical speed.

11. In formula (2), E(Vi) is introduced as the number of expansion nodes. Does this mean how many nodes will be considered around an obstacle? Wasn't the main question determining the side length of an expansion node rather than their number?

12. In formula (2), no need to use the word "default".

13. Considering my comments above about formula (2), this formula should be redefined and show more clearness.

14. Figure 2 is unclear.

15. Algorithm one still is not clearly explaining the idea. Maybe a flowchart can be a better representation.

16. The experimental results (figures and discussions) of autonomous navigation testing are missing. How the experiment robot has found its collision-free path toward the goal location using the proposed algorithm?

17. Figure 12 does not completely match Table 9.

18. The theoretical part of the work must be developed significantly.

Reviewer #2: The authors have addressed all the comments given. I would suggest the authors to add some statistical analysis on the results obtained.

7. PLOS authors have the option to publish the peer review history of their article (what does this mean?). If published, this will include your full peer review and any attached files.

Reviewer #1: No

Reviewer #2: No

---

## [Author Response · Author response to Decision Letter 1]

12 Dec 2021

Dear reviewer,

Thank you for your valuable comments and suggestions concerning our manuscript. The comments are very helpful for revising and improving the paper, as well as the important guiding significance to our researches. We have studied comments carefully and have made corrections. We hope you find the revised manuscript acceptable for publication. Below, we have provided answers to the questions raised by the reviewer, along with detailed explanations for your concerns.

Response to Reviewer 1 Comments

Point 1: The English of the manuscript still requires improvement.

Response 1:

We have read the full text of the manuscript and revised some English issues.

Point 2: Line 48, and line 53 need correction.

Response 2:

We have revised the two incorrect sentences.

Line 48: In section 7, the EBS-A* algorithm is transplanted to a robot and verified the effectiveness of the algorithm is verified in the real world.

Revise: In section 7, the EBS-A* algorithm is tested in the robot operating system (ROS) and its performance is verified in the real world.

Line 53: Regarding the classification of the path planning algorithms, path planning algorithms are divided into more than two groups.

Revise: There are multiple classification methods for path planning algorithms.

Point 3: Line 60-61, what do those abbreviations stand for?

Response 3:

GA represents the genetic algorithm, ACO algorithm stands for ant colony optimization algorithm. These two abbreviations are first mentioned in lines 12-13. ANN algorithm is artificial neural network algorithm, SA algorithm is simulated annealing algorithm, all issues have been revised.

Point 4: Very long sentence from line 71 to 75.

Response 4:

We have revised the long sentence.

Revise: A* is used for shortest path evaluation based on the information regarding the obstacles present in the static environment. The shortest path evaluation for the known static environment is a two-level problem. The problem comprises a selection of feasible node pairs and the shortest path evaluation based on the obtained feasible node pairs.

Point 5: the paragraph starting from line 128 and its intentions are unclear.

Response 5:

We have revised the unclear sentence in the manuscript.

Line 128: A constrained A* method is proposed for USVs in a maritime environment.

Revise: A constrained A* method is proposed for unmanned surface vehicles (USVs) in a maritime environment.

Point 6: the paragraph starting from line 152 is missing a strong motivation for this study.

Response 6:

We have added the motivation for this study.

Revise: The existing improved A* algorithm only optimizes efficiency or robustness. At present, there is no improved algorithm with excellent comprehensive performance. Path planning algorithm plays an essential role in the autonomous navigation of mobile robots. Since mobile robots are widely used in the real world, it is very necessary to propose an improved A* algorithm with strong robustness and high efficiency. It has huge application potential and commercial value in the industrial field. 

Point 7: line 156 "The A* algorithm was first proposed and described in detail in [13]," should be double fact-checked. A* has been proposed firstly half a century ago.

Response 7:

 Following your suggestion. We have revised this reference in the manuscript.

Point 8: Line 175 needs correction.

Response 8:

We have corrected the sentence.

In this section, three optimization strategies are proposed to improve the efficiency and robustness of the A* algorithm.

We responded to points 9-14 together.

Point 9: In formula (2), the sub index of E(V_i) should be corrected, i.e., should be without underscore. The * (if denotes multiplication) is redundant and should be removed. Throughout the paper, often the text format of parameters in formulas and their descriptions in the text do not match.

Point 10: The assumption in line 216 does not make sense. Assuming that the side length of the square is equal to the radius of the robot is a very poor assumption since the obstacles can vary in size greatly. Assuming all obstacles are identical in size with the robot is not realistic.

Point 11: In formula (2), Vi is introduced as the current cruising speed of the robot and Vr is the cruising speed of the robot. What is the difference between these two speeds? It looks likes they refer to an identical speed.

Point 12: In formula (2), E(Vi) is introduced as the number of expansion nodes. Does this mean how many nodes will be considered around an obstacle? Wasn't the main question determining the side length of an expansion node rather than their number?

Point 13: In formula (2), no need to use the word "default".

Point 14: Considering my comments above about formula (2), this formula should be redefined and show more clearness.

Response:

We have revised all the problems in point 9-14, the specific content is as follows:

How is the expansion distance automatically decided for different environments? This is a question we must consider. We refer to the robot equivalent model in the robot operating system to discuss the expansion distance. In this manuscript, the following assumptions of the robot and obstacles are made to simplify the model.

The robot model is equivalent to a cylinder in ROS, with a radius of and cruising speed , which satisfies , where is the maximum cruising speed determined by the performance of the robot. is a speed threshold, is the current speed of the robot. When , the expansion distance is only expanded by one node, and when the current speed is greater than , the probability of a collision between the robot and an obstacle increases. In this case, the number of expanded nodes should increase. The obstacle is equivalent to one grid or more square grids. The mapping rule between the robot model and the map is that the robot radius is equal to the length of a grid. The mapping rule between the number of expansion nodes and the speed is that the ratio of the current speed and the speed threshold.

where is the number of expansion nodes. Regarding the selection of the expansion distance magnitude, when , the expansion distance defaults to the radius of the cylinder as the expansion distance. This distance provides a sufficient “collision buffer” for the reliability of the path, and ensure acceptably minimal waste of the physical space the robot travels. The bilateral expansion distances are the size of a robot itself when there are obstacles on both sides. As the speed of the robot increases, the risk of robot collisions will increase. Correspondingly, only expanding the expansion distance can ensure that the robustness of the algorithm does not decrease. When increases, should increase accordingly, otherwise the risk of robot collision will increase. Therefore, the determination of the expansion distance follows a linear relationship with the speed .

Point 15: Figure 2 is unclear.

Response 15:

We have redrawn Figure 2.

Point 16: Algorithm one still is not clearly explaining the idea. Maybe a flowchart can be a better representation.

Response 16:

 We have added a flowchart of the algorithm and the explanation.

The execution process of the EBS-A* algorithm consists of four steps: 1. Performing expansion distance optimization on the A* algorithm. 2. Performing bidirectional search on the algorithm. 3. Generating an unsmooth path. 4. Performing smoothing process to generate a smooth path. The execution process of the proposed algorithm is shown in Fig 6.

Point 17: The experimental results (figures and discussions) of autonomous navigation testing are missing. How the experiment robot has found its collision-free path toward the goal location using the proposed algorithm?

Response 17:

We have added the experimental results and the experiment process.

The start node and the end node are set in the real world. The robot uses the EBS-A* algorithm to plan a collision-free path to the goal position. This experiment process consists of three steps: 

1. EBS-A* algorithm transplantation; 

2. simultaneous localization and mapping (SLAM) test;

3. a robot autonomous navigation test. 

Algorithm transplantation accomplishes writing EBS-A* algorithm into the ROS. The SLAM test built a test map to use the radar on the robot in the real world. The autonomous navigation test verifies the effectiveness of the algorithm in the real world.

In this experiment, we set an application environment for EBS-A* algorithm and carried out the autonomous navigation test. The EBS-A* algorithm was written into a real mobile robot. The experimental result shows that the robot can independently plan a reliable and smooth path and complete the autonomous navigation from the starting node to the goal node. This experiment verifies that the EBS-A* algorithm can be applied to mobile robots and has the potential to be applied to industrial scenarios.

Point 18: Figure 12 does not completely match Table 9.

Response 18:

 We corrected the data in Table 9 and added the explanation to Table 9 and the original Figure 12 (now Figure 13).

In this research, we have tested the efficiency of the traditional A* algorithm and EBS-A* algorithm. In [16], authors also have tested the efficiency of the traditional A* algorithm and the geometric A* algorithm. We can choose the efficiency of the traditional A* algorithm as a benchmark to compare the efficiency of the EBS-A* algorithm and the geometric A* algorithm. As shown in Table 9, the running time of the A* algorithm is 316.334s and the running time of the geometric A* algorithm is 292.142s in [16]. The running time of the traditional A* algorithm is 36.806s and the running time of the EBS-A* algorithm is 9.747s. The histograms of EBS-A* and geometric A* in Fig 13 are the results of proportional calculations.

Point 19: The theoretical part of the work must be developed significantly.

Response 19:

We have added basic theoretical support to the bidirectional search and smoothing part of the manuscript.

Bidirectional search:

The traditional graph search algorithms are not considered the features of the path planning problem, like DFS and BFS. The path is searched by the strategy set beforehand for any problem, and the search process will not be optimized according to the features of the problem. The A* algorithm is developed based on the BFS algorithm. The concept of heuristic is introduced on the basis of the BFS algorithm. The heuristic information is obtained according to the features of the problem, which will guide the search in the optimal direction. Such as speeding up the search process and improving efficiency. The traditional A* algorithm uses Manhattan distance as its heuristic equation. Manhattan distance is defined as follows:

 (3)

Smoothing:

Bezier curve is a space curve and has good geometric properties, which is proposed by the French engineer Pierre Bezier in 1962. Bezier curve is one of the methods used to smooth the path, it has been widely used in computer graphics and computer-aided design. If the control point of the Bezier curve is a convex polygon, that is, the feature polygon is convex, the Bezier curve is also convex, which is one of the advantages of the Bezier curve. Unlike other types of curves, such as cubic splines or polynomials, Bezier curve does not pass through all the data points used to define it. The points used to define Bezier curve are called control points. Polygons that can be drawn from these control points are called Bezier polygons. The turning points are the points where the slope of the curve changes its sign. Bezier curves have fewer turning points so that it is smoother than cubic splines.

Bezier curve has the following properties:

1. Symmetry, the coefficient of the curve is the same as the reciprocal coefficient.

2. Convex hull properties, Bezier curve is always contained in the convex hull of the polygon defined by all control points.

3. End-points properties, the first control point and the last control point on the curve are exactly the start point and the end point of the Bezier curve.

4. Recursion, which means that the coefficient of the Bezier curve satisfies the following formula.

 (4)

The radius of curvature of the Bezier curve varies smoothly from the starting point to the endpoint because of its continuous higher order derivatives. A Bezier curve of degree n is a parametric curve composed of Bernstein basis polynomials of degree and it can be defined as:

 (5)

Where indicates the normalized time variable, represents the coordinate vector of the control point with and being the components corresponding to the X and Y coordinate, respectively, is the Bernestein basis polynomials, which represents the base function in the expression of Bezier curve, and it is defined as follows:

 (6)

The derivatives of Bezier curve are determined by the control points, and the first derivative of a Bezier curve in formula 5. is expressed as in formula 7. Moreover, higher-order derivatives of a Bezier curve can also be calculated.

(7)

In the two-dimensional space, the curvature of a Bezier curve with respect to t is expressed as follows:

(8)

In the path planning problem, Bezier curve is connected to form a smooth path planning for mobile robots.

Response to Reviewer 2 Comments

Point 1: I would suggest the authors to add some statistical analysis on the results obtained.

Response 1: 

We have added some statistical analysis in this manuscript.

As shown in Table 3，the running time of the EBS-A* algorithm is 138.813s and the traditional A* algorithm is 302.467s on a 100×100 map. The efficiency of the EBS-A* algorithm is only 2.17 times that of the A* algorithm. As shown in Table 4, the efficiency of the EBS-A* algorithm is only 2.14 times that of the A* algorithm on randomized 100×100 maps. The statistical results in Table 3 and Table 4 show that the test results of the algorithm on the fixed map and the random map are consistent. The efficiency of the algorithm is reliable. 

But the efficiency of the EBS-A* algorithm is 4.7 times that of the A* algorithm on a 150×150 map. The reason is that there are only a few obstacles in the 100×100 map, but there are dense obstacles in the 150×150 map. In an environment with dense obstacles, the efficiency of the algorithm is higher. These statistical results have also been verified on 200×200 maps. the efficiency of the EBS-A* algorithm is 5.79 times that of the A* algorithm on a 200×200 map.

All the statistical results of the simulation test show that the efficiency of the EBS-A* algorithm is significantly improved compared with the traditional A* algorithm. These results verify the rationality of the algorithm design.

---

## [Decision Letter · Decision Letter 2]

17 Jan 2022

PONE-D-21-28660R2The EBS-A* algorithm: an improved A* algorithm for path planningPLOS ONE

Dear Dr. Wang,

Thank you for submitting your manuscript to PLOS ONE. After careful consideration, we feel that it has merit but does not fully meet PLOS ONE’s publication criteria as it currently stands. Therefore, we invite you to submit a revised version of the manuscript that addresses the points raised during the review process. Please submit your revised manuscript by Mar 03 2022 11:59PM. If you will need more time than this to complete your revisions, please reply to this message or contact the journal office at plosone@plos.org. Please include the following items when submitting your revised manuscript:A rebuttal letter that responds to each point raised by the academic editor and reviewer(s). You should upload this letter as a separate file labeled 'Response to Reviewers'.A marked-up copy of your manuscript that highlights changes made to the original version. You should upload this as a separate file labeled 'Revised Manuscript with Track Changes'.An unmarked version of your revised paper without tracked changes. You should upload this as a separate file labeled 'Manuscript'.If applicable, we recommend that you deposit your laboratory protocols in protocols.io to enhance the reproducibility of your results. Protocols.io assigns your protocol its own identifier (DOI) so that it can be cited independently in the future. For instructions see: https://journals.plos.org/plosone/s/submission-guidelines#loc-laboratory-protocols. Additionally, PLOS ONE offers an option for publishing peer-reviewed Lab Protocol articles, which describe protocols hosted on protocols.io. Read more information on sharing protocols at https://plos.org/protocols?utm_medium=editorial-email&utm_source=authorletters&utm_campaign=protocols.

We look forward to receiving your revised manuscript.

Kind regards,

Yogendra Arya

Academic Editor

PLOS ONE

Journal Requirements:

Additional Editor Comments:

Still minor revision is required.

Reviewers' comments:

Reviewer's Responses to Questions

**Comments to the Author**

1. If the authors have adequately addressed your comments raised in a previous round of review and you feel that this manuscript is now acceptable for publication, you may indicate that here to bypass the “Comments to the Author” section, enter your conflict of interest statement in the “Confidential to Editor” section, and submit your "Accept" recommendation.

Reviewer #1: All comments have been addressed

Reviewer #2: All comments have been addressed

2. Is the manuscript technically sound, and do the data support the conclusions?

Reviewer #1: Partly

Reviewer #2: Yes

3. Has the statistical analysis been performed appropriately and rigorously? 

Reviewer #1: Yes

Reviewer #2: Yes

4. Have the authors made all data underlying the findings in their manuscript fully available?

Reviewer #1: (No Response)

Reviewer #2: Yes

5. Is the manuscript presented in an intelligible fashion and written in standard English?

Reviewer #1: Yes

Reviewer #2: Yes

6. Review Comments to the Author

Reviewer #1: Line 279, what denote xa,xb,ya, and yb? it should be given after the formula.

Starting from line 289, the Bezier curve theory is explained. However, this method, if used to smooth the path, is not applied in Fig. 4 and Fig. 5. Also, the Bezier curve should be derived for a "single-angle turn" (Fig. 4) and "continuous right-angle turns" (Fig. 5) for the n right-angle turn case.

The paths generated by the proposed algorithm do not show the Bezier curve's promised smoothness, for example in Fig. 7.

Reviewer #2: I am satisfied with all the corrections done by the authors. Hence, I accept the manuscript to be published with PLOSOne

7. PLOS authors have the option to publish the peer review history of their article (what does this mean?). If published, this will include your full peer review and any attached files.

Reviewer #1: No

Reviewer #2: No

---

## [Author Response · Author response to Decision Letter 2]

18 Jan 2022

Dear reviewer,

Thank you for your valuable comments and suggestions concerning our manuscript. The comments are very helpful for revising and improving the paper, as well as the important guiding significance to our researches. We have studied comments carefully and have made corrections. We hope you find the revised manuscript acceptable for publication. Below, we have provided answers to the questions raised by the reviewer, along with detailed explanations for your concerns.

Response to Reviewer 1 Comments

Point 1: Line 279, what denote xa,xb,ya, and yb? it should be given after the formula.

Response 1:

h(n)=|x_a-x_b |+|y_a-y_b | (3)

We have added the statements in this manuscript.

The heuristic function is the minimum cost evaluation value of the A* algorithm from any node to the goal node and helps to reduce the number of nodes traversed. The heuristic function of the traditional A* algorithm uses the Manhattan distance. 

In formula (3), h(n) is the heuristic function, (x_a,y_a) is the coordinates of the goal node, (x_b,y_b) is the coordinates of any node. 

In addition, we have checked the full text of the manuscript to ensure that there are no such issues.

Point 2: Starting from line 289, the Bezier curve theory is explained. However, this method, if used to smooth the path, is not applied in Fig. 4 and Fig. 5. Also, the Bezier curve should be derived for a "single-angle turn" (Fig. 4) and "continuous right-angle turns" (Fig. 5) for the n right-angle turn case.

Response 2:

We have replaced the Fig. 4 and Fig. 5. 

They are as follows:

Fig. 4 Smoothing optimization strategy for a single right-angle turn

Fig. 5 Smoothing optimization strategy for continuous right-angle turns

In Fig. 4, we first decompose a single right-angle turn into two acute angles, and two 45^°angles are generated in the path, as shown in the right picture of Fig. 4 in the original manuscript. We perform Bezier curve smoothing on the 45^° angles, which ultimately generates the right picture of Fig. 4. The smoothing process of Fig. 5 is similar to that of Fig.4.

The Bezier curve for a single right-angle turn is a second-order Bezier curve, its formula is:

B(t)=〖(1-t)〗^2 P_0+2t(1-t) P_1+t^2 P_2,t∈[0,1]

P_0 is the start node, P_1 is the control node, P_2 is the end node, and t is time.

A line segment described by a continuous point Q_0 from P_0 to P_1. A line segment is described by a continuous point Q_1 from P_1 to P_2. The continuous node B(t) from Q_0 to Q_1 describes a second-order Bezier curve. 

The Bezier curve for continuous right-angle turns is a third-order Bezier curve, its formula is:

B(t)=〖(1-t)〗^3 P_0+3t〖(1-t)〗^2 P_1+ 3t^2 (1-t)P_2+ t^3 P_3,t∈[0,1]

P_0 is the start node, P_1 and P_2 are the control nodes, and P_3 is the end node. The curve passes through the control nodes at both ends, namely:

P(0)=P_0

P(1)=P_3 P^' (0)=3(P_1-P_0)

Point 3: The paths generated by the proposed algorithm do not show the Bezier curve's promised smoothness, for example in Fig. 7.

Response 3:

In this manuscript, we apply the proposed algorithm to perform simulation tests on 50×50 maps, 100×100 maps, 150×150 maps and 200×200 maps. In these maps, each grid represents a pixel. The resolution of the test maps is relatively low. Each pixel is a square. The process of path planning is based on pixel points, and the paths generated by the proposed algorithm are composed of pixel points, therefore, the smoothness promised by Bezier curve is not shown in the above maps. In this manuscript, both the theoretical analysis and the real-world case demonstrate the excellent smoothness of the proposed algorithm, as shown in Fig. 16.

We have made all data underlying the findings in our manuscript fully available on GitHub.

---

## [Decision Letter · Decision Letter 3]

28 Jan 2022

The EBS-A* algorithm: an improved A* algorithm for path planning

PONE-D-21-28660R3

Dear Dr. Wang,

We’re pleased to inform you that your manuscript has been judged scientifically suitable for publication and will be formally accepted for publication once it meets all outstanding technical requirements.

Kind regards,

Yogendra Arya

Academic Editor

PLOS ONE

Additional Editor Comments (optional):

The paper is now in acceptable form.

Reviewers' comments:

Reviewer's Responses to Questions

**Comments to the Author**

1. If the authors have adequately addressed your comments raised in a previous round of review and you feel that this manuscript is now acceptable for publication, you may indicate that here to bypass the “Comments to the Author” section, enter your conflict of interest statement in the “Confidential to Editor” section, and submit your "Accept" recommendation.

Reviewer #1: All comments have been addressed

2. Is the manuscript technically sound, and do the data support the conclusions?

Reviewer #1: Yes

3. Has the statistical analysis been performed appropriately and rigorously? 

Reviewer #1: Yes

4. Have the authors made all data underlying the findings in their manuscript fully available?

Reviewer #1: Yes

5. Is the manuscript presented in an intelligible fashion and written in standard English?

Reviewer #1: Yes

6. Review Comments to the Author

Reviewer #1: (No Response)

7. PLOS authors have the option to publish the peer review history of their article (what does this mean?). If published, this will include your full peer review and any attached files.

Reviewer #1: No

---

## [Editor Report · Acceptance letter]

4 Feb 2022

PONE-D-21-28660R3 

The EBS-A* algorithm: an improved A* algorithm for path planning  

Dear Dr. Wang:

I'm pleased to inform you that your manuscript has been deemed suitable for publication in PLOS ONE. Congratulations! Your manuscript is now with our production department. 

Kind regards, 

on behalf of

Dr. Yogendra Arya 

Academic Editor

PLOS ONE